# GGBall: Graph Generative Model on Poincaré Ball

**Tianci Bu**[1,2*] **Chuanrui Wang**[1*] **Hao Ma**[1] **Haoren Zheng**[1] **Xin Lu**[2†] **Tailin Wu**[1†]

[1]Westlake University [2]National University of Defense Technology

`{butianci, wangchuanrui, mahao, zhenghaoren, wutailin}@westlake.edu.cn`

## Abstract

Generating graphs with hierarchical structures remains a fundamental challenge due to the limitations of Euclidean geometry in capturing exponential complexity. Here we introduce **GGBall**, a novel hyperbolic framework for graph generation that integrates geometric inductive biases with modern generative paradigms. GG-Ball combines a Hyperbolic Vector-Quantized Autoencoder (HVQVAE) with a Riemannian flow matching prior defined via closed-form geodesics. This design enables flow-based priors to model complex latent distributions, while vector quantization helps preserve the curvature-aware structure of the hyperbolic space. We further develop a suite of hyperbolic GNN and Transformer layers that operate entirely within the manifold, ensuring stability and scalability. Empirically, GGBall establishes a new state-of-the-art across diverse benchmarks. On hierarchical graph datasets, it reduces the average generation error by up to 18% compared to the strongest baselines. These results highlight the potential of hyperbolic geometry as a powerful foundation for the generative modeling of complex, structured, and hierarchical data domains. Code is available at: `https://github.com/AI4Science-WestlakeU/GGBall`.

## 1 Introduction

Graph generation plays a central role in many scientific and engineering domains, including molecular design, material discovery, and social network modeling (Miller et al., 2024; Shi et al., 2020; Reiser et al., 2022; Luo et al., 2021; Wang et al., 2021a). Recent advances in deep generative models have enabled powerful data-driven approaches to this task. Most existing models operate directly in the discrete graph space, where generation proceeds by sequentially constructing or refining nodes and edges. For instance, diffusion-based models such as GDSS (Jo et al., 2022) and DiGress (Vignac et al., 2023) iteratively denoise graphs in a discrete space, while autoregressive models like GraphRNN (You et al., 2018) and GRAN (Liao et al., 2019) build graphs one node or edge at a time, modeling structural dependencies step-by-step. These approaches offer fine-grained control and explicitly model structural dependencies in the discrete graph domain.

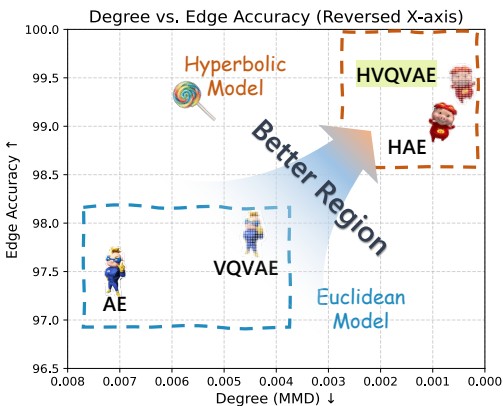

Figure 1: Degree similarity and edge reconstruction accuracy on reconstructed dataset. Hyperbolic models consistently outperform Euclidean baselines.

While these models have shown promise, graph generation remains challenging due to the discrete, combinatorial, and often hierarchical nature of graph data (Guo & Zhao, 2022). To address these issues, latent space generation has emerged as a scalable and flexible alternative. By encoding

---

*Equal contribution, listed in alphabetical order. †Corresponding author.

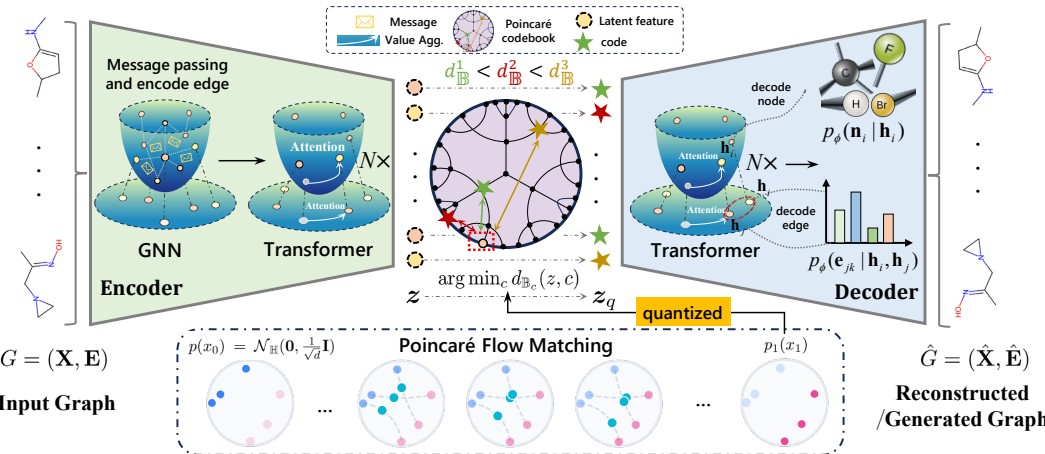

Figure 2: **Overview of our hyperbolic graph generation framework.** We encode graphs into a hyperbolic latent space using a Poincaré GNN and geodesic-attention Transformer. The latent representations are quantized via a Poincaré codebook and modeled with a Poincaré flow prior. A hyperbolic Transformer then decodes the latent code to reconstruct or generate graphs, enabling structure-aware generation in non-Euclidean geometry.

graphs into continuous latent representations and decoding from this space, methods like Graph-VAE (Simonovsky & Komodakis, 2018) and VQGAE (Boget et al., 2023) enable efficient one-shot generation[1]. Nevertheless, these models typically rely on Euclidean latent spaces, which are ill-suited for capturing the hierarchical and compositional nature of real-world graphs, such as community structures and power-law degree distributions (Krioukov et al., 2010).

This geometric mismatch motivates our shift to hyperbolic space, a natural framework for hierarchical representation learning (Mathieu et al., 2019). Unlike Euclidean embeddings that distort parent-child relationships, hyperbolic geometry intrinsically preserves graph hierarchies through its exponentially expanding volume (Chami et al., 2019; Ganea et al., 2018; Krioukov et al., 2010; Sarkar, 2011). As demonstrated in Figure 1, hyperbolic latent models (*e.g.*, HAE, HVQVAE) achieve superior alignment with power-law degree distributions ($4\times$ lower MMD) and higher edge reconstruction accuracy compared to Euclidean counterparts on Community dataset.

Building on these insights, we present **GGBall**, the first Graph Generation framework fully built upon the Poincaré Ball model of hyperbolic space. Our core innovation is a unified, node-only latent representation. This holistic design ensures that the entire graph topology, both nodes and their relational structure, is governed by a single, consistent geometric framework, treating edge connectivity as an emergent property of the latent node geometry. This approach differs from discrete space or hybrid models (Wen et al., 2024) that assign nodes and edges to separate latent spaces. Specifically, we convert a standard Euclidean latent generative pipeline (Rombach et al., 2022) into a fully hyperbolic one: For the encoder, we design a Hyperbolic Vector-Quantized Autoencoder (HVQVAE) that captures graph structure via discrete tokens in the Poincaré ball, initialized through geodesic clustering and optimized using Riemannian methods. For the latent generative process, we leverage flow matching in hyperbolic space to model expressive priors without relying on predefined noise. For architectural support, we develop a modular suite of hyperbolic GNN and Transformer layers that operate entirely within the manifold, ensuring numerical stability and scalability.

Our method achieves state-of-the-art performance across diverse graph generation benchmarks. On hierarchical datasets like COMMUNITY-SMALL and EGO-SMALL, GGBall reduces the average generation error by up to 18% compared to the strongest competitors, highlighting its strength in modeling modular, tree-like structures. On molecular graphs (QM9), GGBall also delivers state-of-the-art results, achieving a best-in-class novelty of 93.77% while maintaining high chemical validity, culminating in the highest overall V.U.N. score. This demonstrates its robust ability to generate diverse and chemically plausible molecules.

---

[1]For more related work, please refer to Appendix C.

## 2 PRELIMINARIES AND PROBLEM DEFINITION

To support our hyperbolic generative framework, we briefly review the Poincaré ball model as the underlying latent space in Section 2.1, followed by the formulation of graph generation in non-Euclidean geometry in Section 2.2. For clarity, all hyperbolic-geometry operators are fully defined in the Appendix E.5.

### 2.1 HYPERBOLIC GEOMETRY

#### 2.1.1 RIEMANNIAN MANIFOLD

A Riemannian manifold $(\mathcal{M}, \mathfrak{g})$ is a smooth manifold equipped with a Riemannian metric tensor field $\mathfrak{g}$, which smoothly assigns to each point $x \in \mathcal{M}$ an inner product $\mathfrak{g}_x : \mathcal{T}_x\mathcal{M} \times \mathcal{T}_x\mathcal{M} \to \mathbb{R}$ on its tangent space $\mathcal{T}_x\mathcal{M}$ (Gallot et al., 1990). For a $n$-dimensional manifold, each tangent space $\mathcal{T}_x\mathcal{M}$ is locally isomorphic to $\mathbb{R}^n$, providing a linear approximation of the manifold at $x$.

#### 2.1.2 HYPERBOLIC SPACE AND POINCARÉ BALL MODEL

Hyperbolic space is a Riemannian manifold of constant negative curvature $-c$, offering a geometric framework for hierarchical data representation. Among its isomorphic models, the Poincaré ball model $(\mathbb{B}_c^n, \mathfrak{g}_c)$ is widely adopted in machine learning due to its conformal structure and numerical stability. Here, $\mathbb{B}_c^n = \{\boldsymbol{x} \in \mathbb{R}^n \mid c\|\boldsymbol{x}\|^2 < 1\}$ defines an open ball of radius $1/\sqrt{c}$, and the metric tensor $\mathfrak{g}_c^x = (\lambda_c^x)^2\mathbf{I}_n$ scales Euclidean distances by the conformal factor $\lambda_c^x = 2(1 - c\|\boldsymbol{x}\|^2)^{-1}$. This induces a Riemannian inner product $\langle \boldsymbol{u}, \boldsymbol{v} \rangle_c^x = (\lambda_c^x)^2\langle \boldsymbol{u}, \boldsymbol{v} \rangle$ for $\boldsymbol{u}, \boldsymbol{v} \in \mathcal{T}_x\mathbb{B}_c^n$ (Ungar, 2008). To enable algebraic operations on hyperbolic coordinates, the Möbius gyrovector framework extends vector space axioms to gyrovectors. The basic binary operation is denoted as the Möbius addition $\oplus_c : \mathbb{B}_c^n \times \mathbb{B}_c^n \to \mathbb{B}_c^n$, which is a noncommutaive and nonassociative addition. We provide a detailed introduction of basic operations in Appendix E.5 and E.6.

#### 2.1.3 GRAPHS IN HYPERBOLIC SPACE

Hyperbolic space is well-suited for embedding graphs with hierarchical or tree-like structures, thanks to its exponential volume growth and negative curvature. Prior works have shown that hyperbolic embeddings better preserve hierarchical relationships and long-range dependencies than their Euclidean counterparts (Nickel & Kiela, 2017; Chami et al., 2019; Krioukov et al., 2010). This is further supported by Gromov's approximation theorem, which states that tree-like structures admit approximate embeddings into hyperbolic space, formally linking hyperbolic geometry to tree-like structures (Appendix E.3).

While these results motivate hyperbolic embeddings for representation learning, graph generation in hyperbolic space remains largely unexplored. Our work addresses this gap by introducing a generative framework that fully exploits curvature-aware geometric priors.

### 2.2 GRAPH GENERATION IN HYPERBOLIC SPACE

In this section, we formalise the task of graph generation in hyperbolic latent space and outline the core idea of our approach.

**Problem definition.** We consider an undirected graph $G = (\mathbf{X}, \mathbf{E})$. Node attributes are represented as one-hot vectors, $\boldsymbol{x} = (\boldsymbol{x}_1, \boldsymbol{x}_2, \ldots, \boldsymbol{x}_m)^T \in \mathbb{R}^{m \times k_1}$, and edges types are represented by $\boldsymbol{e} \in \mathbb{R}^{m \times m \times k_2}$ in such dense matrix representation. The absence of edges is treated as an additional edge type. We use $m$ to denote the total number of nodes in a single graph, $k_1$ and $k_2$ as the number of classes for nodes and edges. The goal of graph generative model is to learn a distribution that can produce graphs whose structure and attributes match those observed in the training set.

**Latent factorization.** Motivated by the insight that hyperbolic space inherently captures edge information through geometry, our idea is to encode edge structure directly into node embeddings. Therefore, we assume that we can introduce a set of hyperbolic latent variables $\boldsymbol{z} = (\boldsymbol{z}_1, \boldsymbol{z}_2, \ldots, \boldsymbol{z}_m), \boldsymbol{z}_i \in$

$\mathbb{B}_c^n$ and factorize the data likelihood as

$$p_\theta(\boldsymbol{x}, \boldsymbol{e}) = \int_{\mathbb{B}_c^n \times \mathbb{B}_c^n \times \cdots \times \mathbb{B}_c^n} p_\theta(\boldsymbol{z}) \, p_\theta(\boldsymbol{x}, \boldsymbol{e} \mid \boldsymbol{z}) \, \mathrm{d}\boldsymbol{z} \tag{1}$$

**Two-stage paradigm.** Inspired by the two-stage generation pipeline in image generation tasks (Rombach et al., 2022), we propose to generate graphs via two sequential steps: (i) Sample a set of node embeddings $\boldsymbol{z}$ that lie on a $n$-dimensional Poincaré ball; (ii) Decode these embeddings into discrete node and edge attributes conditionally as $p_\theta(\boldsymbol{x}, \boldsymbol{e} \mid \boldsymbol{z}) = \prod_{i=1}^m p_\theta(\boldsymbol{x}_i \mid \boldsymbol{z}_i) \prod_{j=1}^m \prod_{k=1}^m p_\theta(\boldsymbol{e}_{jk} \mid \boldsymbol{z}_j, \boldsymbol{z}_k)$, where node labels depend solely on their own latent vectors and edge types depend only on hyperbolic pairwise relations. This conditional independence assumption reduces decoding complexity while fully leveraging hyperbolic distances when modelling edge likelihoods.

## 3 METHOD

We propose a fully hyperbolic framework operating within the Poincaré ball to address the geometric mismatch between graph topology and Euclidean latent spaces mentioned before. First, we propose several basic Poincaré network architecture in Section 3.1, then detail the training paradigm for encoder-decoder framework between graph space and hyperbolic space in Section 3.2. Section 3.3 further describes the modeling of latent prior with normalising flows. The overall architecture is illustrated in Figure 2.

### 3.1 POINCARÉ NETWORK ARCHITECTURE

#### 3.1.1 POINCARÉ GRAPH NEURAL NETWORK

As illustrated in the GNN module of Figure 2, our goal is to construct a hyperbolic message passing that *encodes edge and node information directly into node representation*. The key challenge lies in preserving hyperbolic structure during neighborhood aggregation. Our solution leverages two principles: (1) Adaptive modulation of features based on hyperbolic distances to maintain curvature-aware scaling, and (2) Projection-free operations using closed-form Möbius additions.

Given node embeddings and edge embeddings $\boldsymbol{h}_i, \boldsymbol{h}_{ij} \in \mathbb{B}_c^n$, each layer computes updated features as follows.

**Tangent Space Aggregation.** To enable stable aggregation in curved space, we project neighboring nodes and edge embeddings to the tangent space at the origin using $\log_0^c(\cdot)$, perform Euclidean-style operations, and map the result back via $\exp_0^c(\cdot)$. This yields the following update for node $i$:

$$\boldsymbol{m}_i^{l+1} = \sum_{j \in \mathcal{N}(i)} \mathcal{W}_e \left[ \log_0^c(\boldsymbol{h}_i^l), \log_0^c(\boldsymbol{h}_j^l), \log_0^c(\boldsymbol{h}_{ij}^l) \right], \tag{2}$$

$$\boldsymbol{h}_i^{l+1} = \exp_0^c \left( \log_0^c(\boldsymbol{h}_i^l) + \mathcal{W}_x \left[ \log_0^c(\boldsymbol{h}_i^l), \log_0^c(\mathrm{M}(\boldsymbol{m}_i^{l+1})) \right] \right), \tag{3}$$

where $\mathcal{W}_e, \mathcal{W}_x$ are learned weight functions. This aggregation scheme preserves hyperbolic geometry throughout the message-passing process.

**Distance-Modulated Message Function.** The message function $\mathrm{M}(\cdot)$ integrates node and edge information by modulating the aggregated messages using parameters derived from hyperbolic distances. Specifically, for each edge $(i, j)$, we compute scale and shift coefficients $\gamma_{ij}, \beta_{ij}$ as functions of $d_c(\boldsymbol{h}_i, \boldsymbol{h}_j)$, and apply them to the message: $\mathrm{M}(\boldsymbol{m}_{ij}) = \gamma_{ij} \cdot \boldsymbol{m}_{ij} + \beta_{ij}$.

This curvature-aware modulation allows the model to encode edge strength and structural hierarchy directly into node representations. By design, this mechanism inherently preserves tree-like topologies, unlike Euclidean GNNs that often distort long-range relationships due to flat-space aggregation.

#### 3.1.2 POINCARÉ DIFFUSION TRANSFORMER

After obtaining node representations from the hyperbolic GNN, to further model global graph structure, we adapt diffusion transformers to hyperbolic space by aligning self-attention mechanisms

with geometric priors. The core innovation lies in replacing dot-product attention with geodesic distance scoring and Möbius gyromidpoints to aggregate value, which respects the exponential growth of relational capacity in hyperbolic space. Each layer computes:

**Geodesic Attention.** Score interactions uses hyperbolic distances rather than dot products:

$$\alpha_{ij} \propto \exp(-\tau d_c(\boldsymbol{q}_i, \boldsymbol{k}_j)), \tag{4}$$

where $\boldsymbol{q}_i, \boldsymbol{k}_j$ is projected by input features using Poincaré linear layers (Eq. 16). Values $\boldsymbol{v}_j$ are aggregated using Möbius gyromidpoints to maintain geometric consistency:

$$\boldsymbol{Z}_i = \sum_{j=1}^{T} [\boldsymbol{v}_j, \alpha_{ij}]_c := \tfrac{1}{2} \otimes_c \left( \frac{\sum_j \alpha_{ij} \lambda_c^{\boldsymbol{v}_j} \boldsymbol{v}_j}{\sum_j |\alpha_{ij}| (\lambda_c^{\boldsymbol{v}_j} - 1)} \right) \tag{5}$$

Our Möbius gyromidpoint aggregation can be interpreted as a weighted Fréchet mean on the Poincaré ball, ensuring geometric consistency while being computationally efficient.

**Time-Conditioned Modulation.** We extend the Poincaré transformer with adaptive time-conditioned modulation. Each layer injects timestep embeddings $\boldsymbol{t}_{emb} \in \mathbb{R}^n$ through Euclidean affine transformations of normalized features.

The transformer block maintains hyperbolic consistency via: (1) Multi-head attention splits features using $\beta$-scaling followed by hyperbolic concatenation (Appendix E.6.3). (2) Residual connections employ Möbius addition $\oplus_c$ instead of standard summation. (3) Layer normalization and feed-forward networks operate in tangent space via $\log_0^c / \exp_0^c$ projections (Eq. 17).

## 3.2 Representation Learning in Hyperbolic Latent Space

Building upon the hyperbolic GNN and Transformer layers, we introduce a hyperbolic autoencoding framework for learning graph representations in non-Euclidean latent spaces. As illustrated in Figure 2, this framework employs an encoder-decoder architecture operating entirely within the Poincaré ball model of hyperbolic space.

We explore three architectural variants: (1) a continuous Hyperbolic Graph Autoencoder (HGAE), (2) a probabilistic Hyperbolic Variational Autoencoder (HVAE), and (3) a discrete Hyperbolic Vector-Quantized Autoencoder (HVQVAE). While HVAE is a natural extension of its Euclidean counterpart, we found its Kullback-Leibler (KL) divergence term to be numerically unstable on curved manifolds, often leading to degraded reconstruction quality (see Appendix F.1). In contrast, HVQVAE provides a stable and expressive alternative through latent space discretization, making it our primary model for the full generative framework.

### 3.2.1 Hyperbolic Autoencoder Learning

The Hyperbolic Graph Autoencoder (HGAE) maps graph nodes into hyperbolic space using a Poincaré ball encoder and reconstructs graphs via geometry-aware decoding.

**Encoder & Decoder Design.** The encoder enriches node features with spectral graph properties to capture global topology (Beaini et al., 2021; Vignac et al., 2023; Xu et al., 2019). These features are processed by Euclidean MLPs and projected onto the Poincaré ball via exponential mapping. Subsequent hyperbolic GNN layers aggregate local structural patterns, while stacked hyperbolic transformers propagate global dependencies, finally obtaining node-level representations $\boldsymbol{z}_i$.

The decoder reconstructs node and edge attributes using intrinsic hyperbolic geometry, while edge connectivity and features are conditionally dependent on node pairs. This factorization yields the joint reconstruction probability: $p_\theta(\boldsymbol{x}, \boldsymbol{e} \mid \boldsymbol{h}) = \prod_{i=1}^{n} p_\theta(\boldsymbol{x}_i \mid \boldsymbol{h}_i) \prod_{j=1}^{n} \prod_{k=1}^{n} p_\theta(\boldsymbol{e}_{jk} \mid \boldsymbol{h}_j, \boldsymbol{h}_k)$. Node attributes are predicted by projecting hyperbolic embeddings to the tangent space and applying an MLP. For edge reconstruction, we compute pairwise geometric features:

$$\boldsymbol{f}_{ij} = \left[ \log_{\boldsymbol{0}}^c(\boldsymbol{h}_i), \log_{\boldsymbol{0}}^c(\boldsymbol{h}_j), \log_{\boldsymbol{h}_i}^c(\boldsymbol{h}_j), d_c(\boldsymbol{h}_i, \boldsymbol{h}_j), \cos \theta_{ij} \right], \tag{6}$$

where $d_c$ measures hierarchical distance, $\theta_{ij}$ captures angular relationships, and logarithmic maps encode directional dependencies. These features are decoded into edge probabilities via MLPs.

**Optimization.** The model is trained by minimizing a composite loss function designed to balance reconstruction fidelity with the generation of structurally valid graphs. The overall objective $\mathcal{L}_{AE}$ combines three key components:

$$\mathcal{L} = \mathcal{L}_{recon} + \lambda_{degree}\mathcal{L}_{degree} + \lambda_{reg}\mathcal{L}_{reg} \tag{7}$$

where each term serves a distinct purpose: (1) The reconstruction loss $\mathcal{L}_{recon}$ serves as the primary objective and consists of standard cross-entropy terms for predicting node and edge types from the hyperbolic latent embeddings. (2) A central component of our method is the degree–edge consistency loss $\mathcal{L}_{degree}$, which ensures that the expected degree implied by the edge probabilities is consistent with both the ground-truth degree and the degree predicted at the node level. This encourages the decoder to generate edge patterns that do not contradict local structural roles. The full formulation is provided in Appendix E.7.2. (3) Finally, to regularize the latent space, we include an $\ell_2$ norm penalty $\mathcal{L}_{reg}$ over latent representations.

### 3.2.2 HYPERBOLIC VECTOR-QUANTIZED VARIATIONAL AUTOENCODER LEARNING

To enhance latent space interpretability and expressiveness, we extend HGAE with hyperbolic vector quantization (HVQVAE), which discretizes embeddings into a learnable codebook $\mathcal{C} \in \mathbb{B}_c^n$ and uses a Riemannian optimizer for optimization.

**Codebook Quantization.** Codebook vectors are first initialized via hyperbolic $k$-means clustering (Alg. 16). Each node-level latent representation $z_i$ is then quantized to its nearest codebook entry via $z_q = \arg\min_{c_j \in \mathcal{C}} d_c(z, c_j)$, where $d_c$ denotes the geodesic distance in the Poincaré ball.

**Training Objectives.** The training objective combines three geometrically consistent components:

$$\mathcal{L}_{HVQVAE} = \lambda_1 \mathcal{L}_{AE} + \lambda_2 \underbrace{\mathbb{E}_z[d_c^2(sg(z_q), z)]}_{\text{Commitment}} + \lambda_3 \underbrace{\mathbb{E}_z[d_c^2(z_q, sg(z))]}_{\text{Consistency}}, \tag{8}$$

where $p_\phi$ is the encoder and reconstruction loss is same as HAE; commitment loss anchors latent codes to quantized vectors, and consistency loss updates embeddings via straight-through gradient estimation $sg(\cdot)$.

**Stability Mechanisms.** To prevent codebook collapse, inactive entries are periodically replaced using an expiration threshold. Codebook updates employ weighted Einstein midpoints in the Poincaré ball. More details can be found in Appendix E.7.4.

### 3.3 LATENT DISTRIBUTION MODELING

After training the encoder and decoder, we aim to model a prior over the latent space for generation. Unlike Euclidean spaces, defining stochastic generative processes in hyperbolic geometry is challenging due to the absence of canonical noise and well-defined stochastic dynamics (Fu et al., 2024). To address this, we adopt flow-based models, which offer flexible and deterministic mappings without relying on stochastic processes. While autoregressive model is another choice, it underperforms FM on preliminary experiments, and we leave the exploration for future work.

**Poincaré Flow Matching.** Flow-based generative models (Lipman et al., 2023) define a time-varying vector field $u_t(z_t)$ that generates a probability path $p_t(z_t)$, transitioning between the base distribution $p_0(z_0)$ and the target distribution $p_1(z_1)$. In order to learn the vector field which lies in the tangent space $\mathcal{T}_{z_t}\mathcal{M}$ of $z_t \in \mathbb{B}_c^n$ efficiently, following Chen & Lipman (2024), we minimize the Riemannian conditional flow matching objective:

$$\mathcal{L}_{RCFM} = \mathbb{E}_{t \sim U(0,1), z_0 \sim p(z_0), z_1 \sim p_\phi(X,E), z_t \sim p_t(z_0, z_1)}\|v_\theta(z_t, t) - u_t(z_t|z_1, z_0)\|_{\mathfrak{g}}^2 \tag{9}$$

We define $p(z_0) = \mathcal{N}_{\mathbb{B}}(0, \frac{1}{\sqrt{d}}I)$ (Mathieu et al., 2019) as the prior in hyperbolic space, and $z_1$ as the hyperbolic latent encoding of graph data. The interpolation path is computed via deterministic geodesic interpolation $z_t = \exp_{z_1}(\kappa(t)\log_{z_1}(z_0))$, with $\kappa(t) = 1 - t$.

We parameterize the vector field $v_\theta : \mathbb{B}_c^n \times [0,1] \to \mathcal{T}\mathbb{B}_c^n$ using a Poincaré DiT backbone, followed by a logmap projection to the tangent space. For generation, we integrate on the manifold $\frac{d}{dt}z_t = v_\theta(z_t, t)$ from an initial sample $z_0 \sim p_0$ to obtain $\hat{z}_1$, which is then quantized via the VQ codebook and decoded to obtain generated graphs.

## 4 EXPERIMENT

We organize our experiments around four key questions to evaluate the advantages of hyperbolic latent spaces, focusing on representation quality (Q1, section 4.2), hierarchical structure modeling (Q2, section 4.3), diversity (Q3, section 4.4), and latent space smoothness (Q4, section 4.5):

**Q1:** How well can our model **reconstruct** input graphs from hyperbolic latent representations?
**Q2:** Does the hyperbolic latent facilitate the generation of complex **hierarchical** graph structures?
**Q3:** Does the hyperbolic space provide expressiveness to generate **diverse** structural molecules?
**Q4:** Does our method support smooth and chemically plausible **interpolations** between molecular?

### 4.1 EXPERIMENTAL SETUP

**Baselines.** We compare our methods (**HAE**: hyperbolic autoencoder; **HVQVAE**: vector quantized version; **HVQVAE+Flow**: latent space flow matching) with state-of-the-art models for both generic graph and molecular graph generation. Specifically, the baseline models include: VAE based models, such as GraphVAE (Simonovsky & Komodakis, 2018) and VGAE (Boget et al., 2023), autoregressive based model GraphRNN (You et al., 2018), diffusion models, such as EDP-GNN (Niu et al., 2020), GDSS (Jo et al., 2022), DiGress (Vignac et al., 2023), and the recent HGDM (Wen et al., 2024) that incorporates Poincaré embeddings and edge-specific denoising. Flow-based model including Graph Normalizing Flows (Liu et al., 2019), Graph Autoregressive Flows (Shi et al., 2020), and Categorical Flow matching (Eijkelboom et al., 2024). Please refer to Appendix E.10 for training details.

### 4.2 STUDY ON GRAPH RECONSTRUCTION (Q1)

To answer Q1, we first evaluate the reconstruction fidelity of our autoencoders. This task assesses how effectively the hyperbolic latent space preserves graph information and establishes an upper bound on generative quality, as reconstructions are decoded directly from the latent codes of ground-truth graphs.

As shown in Table 1, our models achieve near-perfect reconstruction. On abstract graphs, HVQVAE consistently improves upon HAE across all MMD metrics and enhances edge accuracy. This trend continues on the QM9 molecular dataset, where HVQVAE significantly boosts chemical validity from 95.18% to 99.14% and overall V.U.N. score from 94.97 to 97.34.

In intuition, compared to HAE, HVQVAE better captures discrete structural motifs (e.g., rings, branches). Compared to HVAE (Appendix F.1), HVQVAE avoids unstable KL regularization in curved manifolds and yields more stable optimization.

Table 1: Reconstruction performance of our baseline models HAE and HVQVAE on abstract graphs (Community-small, Ego-small) and molecular graphs (QM9).

| Model | Community-small | | | | Ego-small | | | |
|---|---|---|---|---|---|---|---|---|
| | Deg. ↓ | Clus. ↓ | Orb. ↓ | Edge Acc. ↑ | Deg. ↓ | Clus. ↓ | Orb. ↓ | Edge Acc. ↑ |
| HAE | 0.0008 | 0.0310 | 0.0007 | 99.10 | 0.0019 | 0.0250 | 0.0048 | 92.40 |
| HVQVAE | **0.0004** | **0.0208** | **0.0005** | **99.39** | **0.0002** | **0.0194** | **0.0018** | **93.20** |

| | QM9 | | | | | |
|---|---|---|---|---|---|---|
| Model | Validity ↑ | Unique ↑ | Novelty ↑ | V.U.N ↑ | Edge Acc. ↑ | Node Acc. ↑ |
| HAE | 95.18 | 99.79 | **100** | 94.97 | 99.48 | 100 |
| HVQVAE | **99.14** | **99.81** | 98.38 | **97.34** | **99.90** | 100 |

### 4.3 GENERIC GRAPH GENERATION (Q2)

**Setup.** Next, we address Q2 by evaluating generative performance on datasets known for their hierarchical and community-based structures: Community-Small and Ego-Small. Following standard protocol (Vignac et al., 2023), we generate a test-set-sized collection of graphs and measure the Maximum Mean Discrepancy (MMD) between the distributions of generated and real graphs across degree, clustering, and orbit statistics.

**Results.** Table 2 shows that our hyperbolic models excel at this task. Our one-shot HVQVAE drastically outperforms its Euclidean counterpart, GraphVAE, reducing the average MMD by 93.3% on Community-small and 86.8% on Ego-small. This demonstrates the powerful inductive bias of the hyperbolic latent space for hierarchical data. When paired with a flow-based prior, our HVQVAE+Flow model sets a new state-of-the-art, achieving the lowest average error on both datasets. This superior performance directly answers Q2, confirming that hyperbolic geometry is highly effective for modeling complex, hierarchical graph structures.

Table 2: Abstract graph generation on Community-small and Ego-small dataset. We evaluate the difference in graph statistics (and their mean) between generated and ground truth graphs. Best results are in **bold**, second best are underlined.

| Type | Space | Method | Community-small | | | | Ego-small | | | |
|---|---|---|---|---|---|---|---|---|---|---|
| | | | Deg.$\downarrow$ | Clus.$\downarrow$ | Orb.$\downarrow$ | Avg.$\downarrow$ | Deg.$\downarrow$ | Clus.$\downarrow$ | Orb.$\downarrow$ | Avg.$\downarrow$ |
| One-shot | $\mathbb{R}^d$ | GraphVAE | 0.3500 | 0.9800 | 0.5400 | 0.6233 | 0.1300 | 0.1700 | 0.0500 | 0.1167 |
| Autoregressive | $\mathbb{G}^d$ | GraphRNN | 0.0800 | 0.1200 | 0.0400 | 0.0800 | 0.0900 | 0.2200 | 0.0030 | 0.1043 |
| | $\mathbb{R}^d$ | VQGAE | 0.0320 | 0.0620 | 0.0046 | 0.0329 | 0.0210 | 0.0410 | 0.0070 | 0.0230 |
| Diffusion | $\mathbb{G}^d$ | EDP-GNN | 0.0530 | 0.1440 | 0.0260 | 0.0743 | 0.0520 | 0.0930 | 0.0070 | 0.0507 |
| | $\mathbb{G}^d$ | GDSS | 0.0450 | 0.0860 | 0.0070 | 0.0460 | 0.0210 | 0.0240 | 0.0070 | 0.0173 |
| | $\mathbb{G}^d$ | DiGress | 0.0470 | **0.0410** | 0.0260 | 0.0380 | - | - | - | - |
| | $\mathbb{G}^d \times \mathbb{B}^n$ | HGDM | 0.0170 | 0.0500 | 0.0050 | 0.0240 | 0.0150 | 0.0230 | 0.0030 | 0.0137 |
| Flow | $\mathbb{G}^d$ | GNF | 0.2000 | 0.2000 | 0.1100 | 0.1700 | 0.0300 | 0.1000 | **0.0010** | 0.0437 |
| | $\mathbb{G}^d$ | GraphAF | 0.0600 | 0.1000 | 0.0150 | 0.0583 | 0.0400 | 0.0400 | 0.0080 | 0.0293 |
| | $\mathbb{G}^d$ | CatFlow | 0.0180 | 0.0860 | 0.0070 | 0.0370 | 0.0130 | 0.0240 | 0.0080 | 0.0150 |
| One-shot | $\mathbb{B}_c^n$ | HVQVAE (ours) | 0.0085 | 0.0681 | 0.0488 | 0.0418 | **0.0071** | 0.0320 | 0.0070 | 0.0154 |
| Flow | $\mathbb{B}_c^n$ | HVQVAE+Flow (ours) | **0.0015** | 0.0589 | **0.0040** | **0.0215** | 0.0133 | **0.0182** | 0.0022 | **0.0112** |

## 4.4 Molecular Graph Generation (Q3)

**Setup.** To answer Q3, we assess our model's ability to generate novel and diverse molecules on the QM9 benchmark. This task challenges the model's expressiveness in a space that is less overtly hierarchical but rich in complex local structures (e.g., rings, bonds). We follow standard protocols for molecular graph generation and reporting chemical validity, uniqueness, and novelty using RDKit.

**Results.** The results in Table 3 highlight a crucial challenge in molecular generation: balancing chemical validity with structural novelty. Many baselines excel at one while sacrificing the other. Our HVQVAE+Flow model resolves this trade-off, simultaneously achieving high validity while generating the most novel molecules (93.77%) of any method. This superior balance leads to state-of-the-art performance on the overall V.U.N. metric (85.34%) and the reliable novelty V.N. score (85.35%). This performance answers Q3 affirmatively, demonstrating that our hyperbolic framework is highly expressive and effective for exploring complex chemical spaces.

Table 3: Molecular graph generation on QM9. We report standard metrics and derived metrics V.N. (Valid × Novel), N/U (Novelty rate). All values are reported as percentages.

| Type | Space | Method | QM9 | | | | | |
|---|---|---|---|---|---|---|---|---|
| | | | Valid.$\uparrow$ | Unique.$\uparrow$ | Novel.$\uparrow$ | V.N.$\uparrow$ | N/U$\uparrow$ | V.U.N.$\uparrow$ |
| One-shot | $\mathbb{R}^d$ | GraphVAE | 45.8 | 30.5 | 66.1 | 30.27 | - | 9.23 |
| Autoregressive | $\mathbb{R}^d$ | VQGAE | 88.6 | - | - | - | - | - |
| Diffusion | $\mathbb{G}^d$ | EDP-GNN | 47.52 | 99.25 | 86.58 | 41.14 | 87.2 | 40.83 |
| | $\mathbb{G}^d$ | GDSS | 95.72 | 98.46 | 86.27 | 82.58 | 87.6 | 81.31 |
| | $\mathbb{G}^d$ | DiGress | 99.0 | 96.2 | 33.4 | 33.07 | 34.7 | 31.81 |
| | $\mathbb{G}^d \times \mathbb{B}^n$ | HGDM | 98.04 | 97.27 | 69.63 | 68.27 | 71.6 | 66.40 |
| Flow | $\mathbb{G}^d$ | GraphAF | 67.00 | 94.51 | 88.83 | 59.52 | **94.0** | 56.25 |
| | $\mathbb{G}^d$ | CatFlow | **99.81** | 99.95 | 49.00 | 48.91 | 49.0 | 48.88 |
| One-shot | $\mathbb{B}_c^n$ | HVQVAE (ours) | 85.93 | 96.59 | 77.64 | 66.72 | 80.4 | 64.44 |
| Flow | $\mathbb{B}_c^n$ | HVQVAE+Flow (ours) | 91.02 | **100.00** | **93.77** | **85.35** | 93.8 | **85.34** |

### 4.5 INTERPOLATION BETWEEN MOLECULES (Q4)

Finally, to address Q4, we examine the smoothness and structure of the latent space via geodesic interpolation. Given two embeddings $\mathbf{h}_0, \mathbf{h}_1 \in \mathbb{B}_c^n$, we trace the geodesic path between them using $\mathbf{h}_t := \exp_{\mathbf{h}_0}\left(t \cdot \log_{\mathbf{h}_0}(\mathbf{h}_1)\right)$ for $t \in [0, 1]$ and decode the intermediate points. A well-structured latent space should yield chemically valid intermediates that blend the properties of the endpoints.

As shown in Figure 3, our method generates smooth and valid molecular transitions. The decoded intermediates exhibit gradual changes in structure and composition, such as ring formation and atom substitution. This high rate of validity along the geodesic path confirms the smoothness of the learned hyperbolic manifold, answering Q4 and suggesting its utility for tasks like property-guided molecule design. We leave the integration of explicit node alignment as a promising direction for future work.

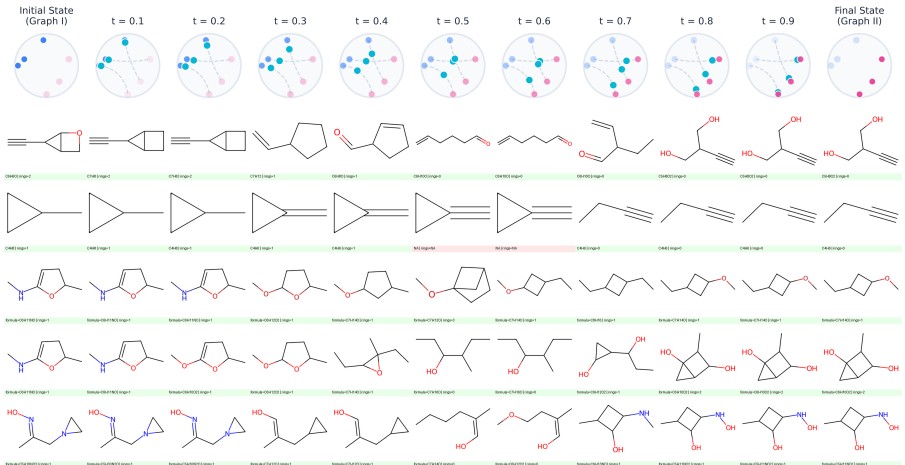

Figure 3: **Geodesic interpolation in hyperbolic latent space between molecular graphs.** Top row shows 10-step latent transitions with 2D-projected node embeddings. Lower rows decode these into molecules, with source and target at ends. Green labels mark valid intermediates; red indicates invalid ones. Chemical formulas and ring counts are annotated.

### 4.6 DISCUSSION

Our experiments validate a powerful, decoupled paradigm for graph generation. The proposed hyperbolic VQ-VAE learns a high-fidelity structural backbone (Q1), enabling not only near-perfect reconstruction but also state-of-the-art generation of both hierarchical abstract graphs (Q2) and complex molecules (Q3), and a smooth latent space for manipulation (Q4), validating this approach as a robust and versatile alternative to end-to-end models.

Furthermore, by operating on a compressed latent representation, our method presents a path towards greater computational efficiency over models that work in the full, high-dimensional graph space (see Appendix G). For **Limitations and Future Work**, please refer to Appendix L.

## 5 CONCLUSION

In this work, we have introduced GGBall, a novel framework for graph generation that leverages hyperbolic geometry to naturally encode the hierarchical structure of complex networks. By combining a vector-quantized autoencoder with Riemannian flow matching in the Poincaré ball, our approach enables curvature-aware and one-shot graph generation. Extensive experiments on both abstract and molecular graph benchmarks validate the effectiveness of our method, showing consistent improvements in reconstruction accuracy and distributional fidelity over Euclidean and graph-space baselines. Our findings highlight the potential of non-Euclidean generative models to accelerate scientific discovery across chemistry, biology, and network science, including applications such as materials design, molecular modeling, and complex graph generation.

## ACKNOWLEDGEMENT

We thank Long Wei, Tao Zhang, Ruiqi Feng, and Tengfei Xu for helpful discussions and valuable feedback on the manuscript. We also gratefully acknowledge the support from the Westlake University Research Center for Industries of the Future and the Westlake University Center for High-Performance Computing. The content of this work is solely the responsibility of the authors and does not necessarily represent the official views of the funding entities.

## ETHICS STATEMENT

This work focuses on developing a generative framework for graph-structured data using synthetic and publicly available datasets, such as Community-Small, Ego-Small, and QM9. All datasets are non-sensitive and do not involve any personally identifiable, medical, or proprietary information. Our method is intended for foundational research in machine learning, materials science, and chemistry, and is not designed for deployment in safety-critical applications without extensive downstream validation. While the model can be applied to molecule generation and scientific discovery, we caution against interpreting the generated outputs as actionable or experimentally validated without proper domain-specific screening. We advocate for careful and responsible use of generative models, especially in sensitive domains such as drug design or social network synthesis.

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

CONTENTS

## A  NOTATION IN THIS PAPER

Table 4: Notations used in this paper.

| Notation | Description |
|----------|-------------|
| $\mathcal{M}$ | Manifold |
| $\mathfrak{g}$ | Metric |
| $\mathbb{R}$ | Real number field |
| $\boldsymbol{x}$ | Point on manifold |
| $c$ | Absolute value of curvature |
| $\mathbb{B}$ | Poincaré ball |
| $h$ | Point on latent space |
| $\langle,\rangle_c$ | Remannian inner product |
| $G$ | A graph |
| $\mathcal{X}$ | Set of graph nodes |
| $\mathcal{E}$ | Set of graph edges |
| $\mathcal{T}$ | Tangent space of a manifold |
| $m$ | Number of nodes |
| $n$ | Number of dimension for a specific space |
| $\oplus_c$ | Möbius Addition |
| $\exp_c$ | Logarithm map |
| $\log_c$ | Exponential map |

## B  THE USE OF LARGE LANGUAGE MODELS (LLMS)

In this paper, we employ LLMs as general-purpose assist tools for text refinement and language polishing. All core research ideas, datasets, and scientific conclusions presented in this paper are our own original contributions.

## C  RELATED WORKS

### C.1  HYPERBOLIC GRAPH REPRESENTATION

**Lorentz Model.**   The Lorentz model represents hyperbolic space as part of a two-sheeted hyperboloid in Minkowski space, enabling efficient geodesic computation and optimization (Busemann, 1955; Ratcliffe, 2006; Epstein & Penner, 1987). It effectively captures hierarchical structures in graphs through continuous embeddings (Nickel & Kiela, 2018), and Lorentz transformations improve knowledge graph embeddings with fewer parameters (Liang et al., 2024). Hyperbolic Graph Convolutional Networks (HGCNs) have shown great potential in capturing hierarchical structures in graphs (Chami et al., 2019). Lorentzian Graph Convolutional Networks (LGCNs) enforce hyperbolic constraints, reducing distortion in tree-like graph representations (Zhang et al., 2021). Further advancements in hyperbolic geometry have enhanced neural architectures for better expressivity and efficiency (Ermolov et al., 2022; Ding et al., 2023; Fu et al., 2024; Yang et al., 2024).

**Poincaré Ball Model.**   The Poincaré ball models hyperbolic space within a unit ball, supporting compact embeddings for hierarchical data (Nickel & Kiela, 2017). Geometry-aware neural layers improve performance in tasks like link prediction and classification (Ganea et al., 2018). MuRP applies Möbius transformations for relation-specific knowledge graph embeddings (Balaževic et al., 2019), and fully convolutional networks capture multi-scale graph features (Lensink et al., 2022). Core components like attention and convolution have been extended to the Poincaré ball for improved efficiency and stability (Shimizu et al., 2020), followed by a fully hyperbolic residual network that enhances robustness to distributional shifts and adversarial perturbations (Van Spengler et al., 2023). Unlike prior work that focuses on discriminative tasks, our framework leverages the Poincaré ball for generative modeling—specifically, learning and sampling discrete graph structures via a hyperbolic latent space. This extends the utility of hyperbolic geometry from representation learning to structured data generation.

## C.2 Graph Generation

**Autoencoder-Based Models.** Autoencoder-based models use latent representations to generate graphs. VAEs (Kingma et al., 2013) and VQVAEs (Van Den Oord et al., 2017; Razavi et al., 2019) are foundational in this area. JT-VAE builds molecular graphs by first generating a tree of substructures, then assembling them with message passing (Jin et al., 2018). Combining VAEs with graph transformers improves scalability and interpretability (Mitton et al., 2021). VQVAEs like VQ-T2G support text-conditioned graph generation by learning discrete latent codes aligned with textual prompts (Walker et al., 2021). Our model extends prior autoencoder-based approaches by using a hyperbolic latent space to better capture hierarchy and improve generative quality.

**Autoregressive Models.** Autoregressive models generate graphs by sequentially adding nodes and edges. GraphRNN models the conditional distribution of graph structures using an RNN, producing diverse graphs (You et al., 2018). GRAN improves efficiency by incorporating attention and generating graph blocks (Liao et al., 2019). G2PT adopts transformer-based next-token prediction for goal-oriented graph generation and property prediction (Chen et al., 2025). In contrast, our method avoids sequential generation in graph space, which requires an explicit ordering, and instead enables one-shot decoding with stronger inductive bias for hierarchical structures.

**Diffusion Models.** Diffusion models are effective for graph generation by transforming noise into structured graphs through a diffusion process. DiGress uses a discrete denoising diffusion model with a graph transformer to handle categorical node and edge attributes, achieving strong results (Vignac et al., 2023). DisCo extends this by modeling diffusion in a discrete-state continuous-time setting, balancing quality and efficiency while preserving graph discreteness (Xu et al., 2024). Concurrent to our work, Wen et al. (2024) proposed the Hyperbolic Graph Diffusion Model (HGDM), which performs diffusion in a hybrid latent space: node embeddings evolve in hyperbolic space while edge embeddings are modeled in Euclidean space, updated by two independent score networks. In contrast, our GGBall framework embeds the entire graph into a compact node-only hyperbolic latent space, implicitly encoding edge structure in the geometry of node embeddings. During generation, we only evolve these node embeddings, with edges reconstructed by the decoder. This unified design eliminates the need for separate latent processes and leads to improved scalability and generative quality.

**Flow Matching.** Flow matching methods learn deterministic transport maps to transform simple distributions into complex graph structures. The Flow Matching framework trains continuous normalizing flows by regressing vector fields along fixed probability paths without simulation (Lipman et al., 2023). DeFoG adapts this to discrete graphs, separating training and sampling for improved flexibility (QIN et al., 2025). CatFlow frames flow matching as variational inference, enabling efficient generation of categorical data (Eijkelboom et al., 2024). In materials science, FlowMM applies Riemannian flow matching to generate high-fidelity materials, showing the method's broad applicability (Miller et al., 2024). Building on these advances, we extend flow matching to a hyperbolic latent space, enabling geometry-aware graph generation with improved alignment to hierarchical structures.

# D  ALGORITHMS

---

**Algorithm 1** Hyperbolic $k$-Means Clustering

---

**Require:** Samples $\{z_i \in \mathbb{B}_c^d\}_{i=1}^N$, cluster count $K$, iterations $T$
**Ensure:** Codebook $\{c_j \in \mathbb{B}_c^d\}_{j=1}^K$, cluster sizes $\{n_j\}_{j=1}^K$
1: Initialize $c_j^{(0)} \leftarrow$ sample_vectors$(\{z_i\}, K)$             ▷ Random initial centers
2: **for** $t = 1$ **to** $T$ **do**
3:      Compute pairwise distances: $d_{ij} = -d_c^2(z_i, c_j^{(t-1)})$    (Eq. 14)
4:      Assign clusters: $b_i \leftarrow \arg\max_j d_{ij}, \ \forall i$
5:      Count clusters: $n_j \leftarrow |\{i : b_i = j\}|, \ \forall j$
6:      **for** $j = 1$ **to** $K$ **do**
7:          **if** $n_j = 0$ **then**
8:             $c_j^{(t)} \leftarrow c_j^{(t-1)}$             ▷ Retain dead clusters
9:          **else**
10:            $c_j^{(t)} \leftarrow$ WeightedMidpoint$(\{z_i\}, \mathbb{I}[b_i = j])$
11:            $c_j^{(t)} \leftarrow \text{proj}_{\mathbb{B}_c^d}(c_j^{(t)})$         ▷ Manifold projection
12:          **end if**
13:      **end for**
14:      **if** $\max_j d_c(c_j^{(t)}, c_j^{(t-1)}) < \epsilon$ **then break**
15:      **end if**
16: **end for**

---

**Algorithm 2** Flow Matching for Graph Generation

---

**Require:** Pretrained VQVAE with hyperbolic latent space, flow-based model
**Ensure:** Generated graph $G_{\text{gen}}$
1: Sample latent codes from the standard normal distribution in hyperbolic space:
2:      $z_0 \sim \mathcal{N}(0, I)$
3: Apply flow transformation to move from the source distribution to the target distribution in the latent space:
4:      $z_t = \text{Flow}_\theta(z_0, t)$     for $t \in [0, 1]$
5: Quantize the latent code to the nearest codebook vector:
6:      $z_q = \arg\min_c d_{\mathbb{B}_c}(z_t, c)$
7: Decode the quantized latent code back to a graph:
8:      $G_{\text{gen}} = D_\psi(z_q)$
9: Return the generated graph $G_{\text{gen}}$

---

---

**Algorithm 3** Hyperbolic Graph Autoencoder (HGAE)

---

**Require:** Graph $G = (\mathbf{X}, \mathbf{E})$, hyperbolic curvature $c$, layers $L$, margins $r_{\min}, r_{\max}$

**Encoding Stage**

1: Compute structural and spectral features $\mathbf{S}$ (*e.g.*, graph Laplacian eigenfeatures).
2: Preprocess attributes using Euclidean MLP:
$$\mathbf{X}_h = \exp_0^c(\text{MLP}([\mathbf{X}, \mathbf{S}])), \quad \mathbf{E}_h = \exp_0^c(\text{MLP}(\mathbf{E}))$$
3: Initialize hyperbolic embeddings: $z_i^{(0)} \leftarrow \text{GNN}_\theta(\mathbf{X}_h, \mathbf{E}_h)$.
4: **for** $l = 1, \ldots, L$ **do**
5:     Perform hyperbolic attention-based message passing:
$$z_i^{(l)} \leftarrow \text{Attention}_\theta(z_i^{(l-1)})$$
6: **end for**
7: Obtain final embeddings: $Z = \{z_i^{(L)}\}$.

**Decoding Stage**

8: **for** each node $i$ **do**
9:     Project hyperbolic embedding to tangent space:
$$\log_0^c(z_i^{(L)}) \in T_0 \mathbb{B}_c^n$$
10:     Predict node attributes:
$$p_\phi(\mathbf{x}_i | z_i^{(L)}) = \text{Softmax}(\text{MLP}(\log_0^c(z_i^{(L)})))$$
11: **end for**
12: **for** each edge $(i, j)$ **do**
13:     Compute hyperbolic geometric features:
$$f_{ij} = [\log_0^c(z_i^{(L)}), \log_0^c(z_j^{(L)}), \log_{z_i^{(L)}}^c(z_j^{(L)}), d_c(z_i^{(L)}, z_j^{(L)}), \cos\theta_{ij}]$$
14:     Predict edge attributes:
$$p_\phi(e_{ij} | z_i^{(L)}, z_j^{(L)}) = \text{Softmax}(\text{MLP}(f_{ij}))$$
15: **end for**
16: **return** Final embeddings $Z$, trained encoder-decoder

---

**Algorithm 4** Hyperbolic Vector Quantized Variational AutoEncoder (HVQVAE)

---

**Require:** Dataset $\mathcal{D}$, codebook size $K$, hyperbolic curvature $c$, EMA parameter $\beta$, margins $r_{\min}, r_{max}$

**Initialization**

1: Initialize hyperbolic codebook $\mathcal{C} = \{c_j\}_{j=1}^K \subset \mathbb{B}_c^d$ using hyperbolic k-means or random sampling.

**Encoding and Quantization**

2: **for** each data point $x \in \mathcal{D}$ **do**
3:     Obtain latent embedding $z = \text{Encoder}_\phi(x) \in \mathbb{B}_c^d$
4:     Quantize embedding to nearest codebook vector using hyperbolic distance:
$$z_q = \arg\min_{c_j \in \mathcal{C}} d_c(z, c_j)$$
5: **end for**

**Decoding**

6: Reconstruct input from quantized embedding: $\hat{x} = \text{Decoder}_\theta(z_q)$

**EMA Codebook Update**

7: **for** each codebook vector $c_j$ **do**
8:     Compute centroid $\mu_j$ from assigned latent vectors $z_i$:
$$\mu_j = \frac{1}{2} \bigoplus_c \left( \frac{\sum_i w_{ij} \lambda_{z_i}^c z_i}{\sum_i |w_{ij}|(\lambda_{z_i}^c - 1)}, \frac{\sum_i w_{ij} z_i}{\sum_i |w_{ij}|} \right)$$
9:     EMA update codebook vector using Einstein midpoint:
$$c_j^{(t+1)} = \text{proj}_{\mathbb{B}_c^d}\left( [c_j^{(t)}, \beta]_c \oplus_c [\mu_j, 1-\beta]_c \right)$$
10: **end for**

**Dead Codes Revival**

11: **if** a codebook vector $c_j$ has not been updated for long periods **then**
12:     Re-initialize codebook vector $c_j$ to a new vector sampled from $\mathbb{B}_c^d$.
13: **end if**
14: **return** Trained encoder, decoder, and hyperbolic codebook

---

---

**Algorithm 5** Training HVQVAE

---

**Require:** Dataset $\{G_i\}$ of graphs
**Ensure:** Trained HVQVAE model on hyperbolic
 1: Initialize hyperbolic encoder $E_\phi$, codebook $\mathcal{C}$, and decoder $D_\psi$
 2: **for** each graph $G_i$ in the dataset **do**
 3:   Map the graph's node and edge features to the hyperbolic space via the Poincaré ball model:
 4:     $Z_i = \text{Exp}_0(E_\phi(G_i))$               $\triangleright$ Exponential map to hyperbolic space
 5:   Apply message passing and attention to encode information in latent space
 6:   Quantize the latent code via the hyperbolic codebook $\mathcal{C}$:
 7:     $z_q = \arg\min_c d_{\mathbb{B}_c}(z, c)$
 8:   Decode the quantized latent codes back to the original graph structure:
 9:     $\hat{G}_i = D_\psi(z_q)$
 10:   Update model parameters using the loss Eq. 40
 11: **end for**

---

## E   ADDITIONAL TECHNICAL DETAILS

### E.1   REMANNIAN MANIFOLD

A Riemannian manifold $(\mathcal{M}, \mathfrak{g})$ is a smooth manifold $\mathcal{M}$ equipped with an inner product $\mathfrak{g}_p : T_p\mathcal{M} \times T_p\mathcal{M} \to \mathbb{R}$ on the tangent space $T_p\mathcal{M}$ at each point $p \in \mathcal{M}$, which varies smoothly with $p$. The inner product $g$ is called the Riemannian metric and induces a notion of distance and angle on the manifold.

Given a point $p \in \mathcal{M}$ and a tangent vector $v \in T_p\mathcal{M}$, the *exponential map* $\exp_p : T_p\mathcal{M} \to \mathcal{M}$ maps $v$ to the point reached by traveling along the geodesic starting at $p$ in the direction of $v$ for unit time. Conversely, the *logarithmic map* $\log_p : \mathcal{M} \to T_p\mathcal{M}$, when defined, maps a point $q \in \mathcal{M}$ back to the tangent vector at $p$ corresponding to the initial velocity of the geodesic from $p$ to $q$. These two maps allow the manifold geometry to be locally approximated using linear operations in the tangent space.

### E.2   HYPERBOLIC GEOMETRY

Let $(\mathcal{M}, \mathfrak{g})$ be a complete, simply connected Riemannian manifold of constant sectional curvature $\kappa < 0$. Then $(\mathcal{M}, \mathfrak{g})$ is called the *n-dimensional hyperbolic space* with curvature $\kappa$, denoted $\mathbb{H}^n_\kappa$. Up to isometry, there exists a unique such space for each dimension $n$ and curvature $\kappa$, making hyperbolic space a canonical example of a negatively curved space form in Riemannian geometry.

In practice, several coordinate models are commonly used to represent this manifold in different ambient geometries. These include the **Lorentz model**, the **Poincaré ball model**, and the **Beltrami–Klein model**. Although their coordinate expressions differ, these models are all isometric to one another and describe the same underlying hyperbolic space. Closed-form mappings exist between the models, allowing flexibility in geometric computation.

We summarize below the Riemannian metrics associated with each of these models.

**Lorentz Model.** This model is realized as a hypersurface embedded in $(n+1)$-dimensional Minkowski space $\mathbb{R}^{n+1}_1$ equipped with the Lorentzian inner product:
$$\langle \mathbf{x}, \mathbf{y} \rangle_\mathcal{L} = \mathbf{x}^\top \mathfrak{g}_\mathcal{L} \mathbf{y}, \quad \text{where} \quad \mathfrak{g}_\mathcal{L} = \text{diag}(-1, 1, \ldots, 1).$$

**Poincaré Ball Model.** Defined on the open ball $\mathbb{B}^n_c = \{\mathbf{x} \in \mathbb{R}^n \mid c\|\mathbf{x}\|^2 < 1\}$ and here $c = -\kappa$. The Poincaré model uses a conformal metric:
$$\mathfrak{g}^c_\mathbf{x} = \lambda^2_\mathbf{x} I_n, \quad \text{with} \quad \lambda_\mathbf{x} = \frac{2}{1 - c\|\mathbf{x}\|^2}.$$

**Beltrami–Klein Model.** Also defined on the same open ball, the Beltrami–Klein model uses a non-conformal metric. Unlike the Poincaré model, it does not preserve angles, but has the advantage that *geodesics are represented as straight lines* within the Euclidean ball:
$$\hat{\mathfrak{g}}^c_\mathbf{x} = (1 - c\|\mathbf{x}\|^2)^{-1} I_n + (1 - c\|\mathbf{x}\|^2)^{-2} c \mathbf{x}\mathbf{x}^\top.$$

These models provide flexible choices for representing hyperbolic geometry depending on the task at hand, including optimization, inference, and geometric embedding in machine learning applications. Poincaré ball here is preferred in graph embedding because it preserves local angles (conformal), naturally encodes hierarchy near the boundary, and supports efficient optimization in a bounded Euclidean space.

### E.3 GRAPHS IN HYPERBOLIC SPACE

Previous works (Sarkar, 2011; Fu et al., 2024) have suggested that graph features are well-suited for embedding in hyperbolic space due to its tree-like structure Coornaert et al. (2006).

**Theorem 1** *(de Gromov's approximation theorem) For any $\delta$-hyperbolic space, there exists a continuous mapping from the space to an $R$-Tree such that the distances between points are approximately preserved with a small error term. More formally, the mapping $\Phi : \mathcal{X} \to T$ satisfies the following distance preservation property:*

$$|x - y| - 2k\delta \le |\Phi(x) - \Phi(y)| \le |x - y|,$$

*where $\mathcal{X}$ is the $\delta$-hyperbolic space, $T$ is the $R$-Tree, and $k$ is the number of sampled points in the mapping.*

### E.4 WHY WE CHOOSE THE POINCARÉ BALL MODEL

**Theoretical Motivation**   We selected the Poincaré Ball model over the Lorentz model for two primary reasons:

- **Numerical Stability:** The Poincaré model is defined on a bounded domain ($\mathbb{B}_c^n$), which prevents latent variable norms from diverging and is inherently more stable for floating-point arithmetic. In contrast, the Lorentz model is unbounded and prone to numerical precision issues in high dimensions, especially for points far from the origin.

- **Geometric Properties:** The Poincaré model is conformal, meaning it preserves local angles. This property benefits learning tasks where local structure is important.

**Practical Comparison**   Furthermore, we conducted a *geometric round-trip experiment* to compare the Poincaré and Lorentz models. Two points $x_0, x_1$ were sampled from a wrapped normal distribution. We then computed the recovered point

$$x_1' = \exp_{x_0} \left( \log_{x_0}(x_1) \right)$$

and measured both the Euclidean L2 error $\|x_1 - x_1'\|_2$ and the manifold distance $d_c(x_1, x_1')$, which ideally should be zero. Results are reported as mean $\pm$ standard deviation over 10 runs.

Table 5: Round-trip numerical stability comparison between the Poincaré Ball and Lorentz models.

| Model | Dim | L2 Error (Mean $\pm$ Std) | Manifold Error (Mean $\pm$ Std) |
|---|---|---|---|
| Poincaré Ball | 8 | $(7.11 \pm 5.59)$e$-16$ | $(4.83 \pm 3.33)$e$-15$ |
| Lorentz | 8 | $(1.64 \pm 3.62)$e$-11$ | $(1.41$e$-4 \pm 1.53$e$-11)$ |
| Poincaré Ball | 32 | $(8.25 \pm 4.01)$e$-16$ | $(4.66 \pm 2.59)$e$-15$ |
| Lorentz | 32 | $126.40 \pm 128.28$ | $0.72 \pm 0.70$ |
| Poincaré Ball | 64 | $(8.86 \pm 5.90)$e$-16$ | $(4.01 \pm 2.82)$e$-15$ |
| Lorentz | 64 | $2958.63 \pm 1546.06$ | $5.95 \pm 0.84$ |

Table 5 shows that the Poincaré Ball model consistently achieves round-trip errors on the order of machine precision ($\sim 10^{-15}$), demonstrating **near-perfect numerical reversibility** across all dimensions. In contrast, the Lorentz model's error **explodes catastrophically** in higher dimensions. For example, at 64 dimensions, the mean L2 error exceeds 2900 with large variance, indicating a complete numerical breakdown.

This provides empirical evidence that the Lorentz model is unsuitable for high-dimensional deep learning applications, validating our choice of the Poincaré Ball model for its robustness and numerical stability.

### E.5 BASIC OPERATIONS ON POINCARÉ BALL

#### E.5.1 MÖBIUS ADDITION AND TRANSFORMATIONS

Möbius addition is defined on Poincaré ball as follows:

$$\boldsymbol{x} \oplus_c \boldsymbol{y} = \frac{\left(1 + 2c\langle \boldsymbol{x}, \boldsymbol{y}\rangle + c\|\boldsymbol{y}\|^2\right)\boldsymbol{x} + \left(1 - c\|\boldsymbol{x}\|^2\right)\boldsymbol{y}}{1 + 2c\langle \boldsymbol{x}, \boldsymbol{y}\rangle + c^2\|\boldsymbol{x}\|^2\|\boldsymbol{y}\|^2}, \tag{10}$$

which reverse calculation is defined by

$$\boldsymbol{x} \ominus_c \boldsymbol{y} = \boldsymbol{x} \oplus_c (-\boldsymbol{y}). \tag{11}$$

This operation is closed on $\mathbb{D}^n$ and defines a non-associative algebraic structure known as a **gyrogroup**.

Unlike Euclidean addition, Möbius addition is not associative. Instead, it satisfies a weaker form of associativity known as **gyroassociativity**:

$$u \oplus (v \oplus_c w) = (u \oplus_c v) \oplus_c \text{gyr}[u, v](w),$$

Accordingly, when considering transformations on the Poincaré disk, it is important to account for the underlying non-associative gyrogroup structure. Throughout this work, all Möbius additions are consistently defined as left additions, which naturally gives rise to operations where composing with the inverse yields the identity transformation.

#### E.5.2 LOGARITHM MAP, EXPONENTIAL MAP, AND GEODESIC DISTANCE

The exponential map $\exp_c^x : \mathcal{T}_x\mathbb{B}_c^n \to \mathbb{B}_c^n$ is described in as follows:

$$\exp_{\boldsymbol{x}}^c(\boldsymbol{v}) = \boldsymbol{x} \oplus_c \frac{1}{\sqrt{c}}\tanh\left(\frac{\sqrt{c}\lambda_{\boldsymbol{x}}^c\|\boldsymbol{v}\|}{2}\right)[\boldsymbol{v}], \quad \forall \boldsymbol{x} \in \mathbb{B}_c^n, \boldsymbol{v} \in \mathcal{T}_{\boldsymbol{x}}\mathbb{B}_c^n. \tag{12}$$

The logarithmic map $\log_c^x = (\exp_c^x)^{-1} : \mathbb{B}_c^n \to \mathcal{T}_x\mathbb{B}_c^n$ provides the inverse operation:

$$\log_{\boldsymbol{x}}^c(\boldsymbol{y}) = \frac{2}{\sqrt{c}\lambda_{\boldsymbol{x}}^c}\tanh^{-1}\left(\sqrt{c}\| \ominus_c \boldsymbol{x} \oplus_c \boldsymbol{y}\|\right)[\ominus_c\boldsymbol{x} \oplus_c \boldsymbol{y}], \quad \forall \boldsymbol{x}, \boldsymbol{y} \in \mathbb{B}_c^n. \tag{13}$$

Geodesic distance between $\boldsymbol{x}, \boldsymbol{y} \in \mathbb{B}_c^n$ is defined as:

$$d_c(\boldsymbol{x}, \boldsymbol{y}) = \frac{2}{\sqrt{c}}\tanh^{-1}\left(\sqrt{c}\| \ominus_c \boldsymbol{x} \oplus_c \boldsymbol{y}\|\right) = \|\log_{\boldsymbol{x}}^c(\boldsymbol{y})\|_{\boldsymbol{x}}^c. \tag{14}$$

Despite the noncommutative aspect of the Möbius addition $\oplus_c$, this distance function becomes commutative thanks to the commutative aspect of the Euclidean norm of the Möbius addition.

### E.6 HYPERBOLIC NETWORK BACKBONE

Prior efforts to construct hyperbolic neural networks predominantly leveraged the **Lorentz model**, yet its numerical instability (particularly the non-invertibility of exponential and logarithmic maps in high dimensions) limits scalability. In contrast, the **Poincaré ball model** offers superior numerical robustness and conformal structure, making it a pragmatic foundation for building deep architectures. Building on these insights, we unify and extend hyperbolic machine learning by developing a comprehensive suite of foundational components and mainstream network modules within the Poincaré framework.

#### E.6.1 POINCARÉ MULTINOMIAL LOGISTIC REGRESSION

Hyperbolic hyperplanes generalize Euclidean decision boundaries by constructing sets of geodesics orthogonal to a tangent vector $\boldsymbol{a} \in \mathcal{T}_p\mathbb{B}_c^n \setminus \{\boldsymbol{0}\}$ at a base point $\boldsymbol{p}$. However, traditional formulations suffer from over-parameterization of the hyperplane. Shimizu et al. (2020) address this by proposing unidirectional hyperplanes, where the base point $\boldsymbol{q}_k$ is constrained as $\boldsymbol{q}_k = \exp_0^c(r_k[\boldsymbol{a}_k])$ using a scalar bias $r_k \in \mathbb{R}$. This reduces redundancy and stabilizes optimization. Building on this, hyperbolic

multinomial logistic regression (MLR) generalizes Euclidean MLR by measuring distances to margin hyperplanes. For $K$ classes, the unidirectional hyperbolic MLR formulation for all $\boldsymbol{x} \in \mathbb{B}_c^n$ is defined as:

$$\boldsymbol{v}_k = p(y = k \mid \boldsymbol{x}) \propto \text{sign}\left(\langle -\boldsymbol{q}_k \oplus_c \boldsymbol{x}, \boldsymbol{a}_k \rangle\right) \|\boldsymbol{a}_k\| d_c\left(\boldsymbol{x}, \bar{H}_{\boldsymbol{a}_k, r_k}^c\right), \quad (15)$$

where $d_c(\boldsymbol{x}, \bar{H}_{\boldsymbol{a}_k, r_k}^c)$ denotes the hyperbolic distance from $\boldsymbol{x}$ to the $k$-th unidirectional hyperplane. $\boldsymbol{a}_k$ and $r_k$ are learnable parameters. This formulation preserves the interpretability of Euclidean margins while leveraging the exponential growth of hyperbolic space for hierarchical classification.

**Poincaré Fully Connected layer.** The Poincaré fully connected (FC) layer generalizes Euclidean affine transformations to hyperbolic space while preserving geometric consistency. Following Shimizu et al. (2020), we adopt a reparameterized formulation that circumvents the parameter redundancy inherent in earlier hyperbolic linear layers (Ganea et al., 2018). For an input $\boldsymbol{x} \in \mathbb{B}_c^n$, the layer computes the output $\boldsymbol{y} \in \mathbb{B}_c^m$ as:

$$\boldsymbol{y} = \mathcal{F}^c(\boldsymbol{x}; \boldsymbol{Z}, \boldsymbol{r}) := \boldsymbol{w}(1 + \sqrt{1 + c\|\boldsymbol{w}\|^2})^{-1}, \text{ where } \boldsymbol{w} := (c^{-\frac{1}{2}} \sinh\left(\sqrt{c}\, \boldsymbol{v}_k(\boldsymbol{x})\right))_{k=1}^m. \quad (16)$$

where $\boldsymbol{v}_k(\boldsymbol{x})$ denotes the signed hyperbolic distance from $\boldsymbol{x}$ to the unidirectional hyperplane $\bar{H}_{\boldsymbol{a}_k, r_k}^c$ which is described in Eq. 15, derived via parallel transport and Möbius addition (Shimizu et al., 2020). This formulation ensures that each output dimension corresponds to a geometrically interpretable distance metric in $\mathbb{B}_c^m$, aligning with the intrinsic curvature of the manifold.

**Poincaré Layernorm.** We introduce Poincaré LayerNorm, a novel normalization module that stabilizes training in hyperbolic networks while preserving geometric integrity. Unlike Euclidean LayerNorm, which operates directly on manifold coordinates, our method leverages the logarithmic map to project hyperbolic features $\boldsymbol{x} \in \mathbb{B}_c^n$ to the tangent space at the origin $\mathcal{T}_0 \mathbb{B}_c^n$. Here, standard LayerNorm is applied to the Euclidean-flattened features, followed by an exponential map to reproject the normalized result back to $\mathbb{B}_c^n$. Formally, for input $\boldsymbol{x}$:

$$\boldsymbol{x}_{norm} = \exp_0^c(\text{LayerNorm}(\log_0^c(\boldsymbol{x}))), \quad (17)$$

where weights and biases in the LayerNorm submodule are initialized to minimize distortion. This approach decouples normalization from manifold curvature, ensuring stable gradient propagation without violating hyperbolic geometry.

**Poincaré ResNet.** For deep hyperbolic architectures, we adapt residual connections to mitigate vanishing gradients, inspired by Chami et al. (2019). Given a hyperbolic feature $\boldsymbol{x} \in \mathbb{B}_c^n$ and a nonlinear transformation $\mathcal{F}(\boldsymbol{x})$, the Poncaré ResNet block computes:

$$\boldsymbol{x}_{out} = \boldsymbol{x} \oplus_c \mathcal{F}(\boldsymbol{x}), \quad (18)$$

### E.6.2 POINCARÉ GRAPH NEURAL NETWORKS

We introduce a Poincaré ball-based graph neural network (GNN) that generalizes message passing to hierarchical graph structures. Unlike Lorentz-based approaches that operate on hyperboloid coordinates, our architecture leverages the conformal structure of the Poincaré ball to enable stable gradient propagation and efficient hyperbolic-linear operations. Let $\boldsymbol{H}_X = \{\boldsymbol{h}_i \in \mathbb{B}_c^n\}_{i=1}^N$ denote node features and $\boldsymbol{H}_E = \{\boldsymbol{h}_{ij} \in \mathbb{B}_c^n\}_{(i,j) \in \mathcal{E}}$ edge features. Our architecture cooperates via two geometrically consistent phases:

**Hyperbolic Message Passing and Edge Update.** For each edge $(i, j)$, compute adaptive shift and scale parameters from the hyperbolic distance $d_c(\boldsymbol{h}_i^l, \boldsymbol{h}_j^l)$:

$$\boldsymbol{\gamma}_{ij}^l, \boldsymbol{\beta}_{ij}^l = \text{Linear}(d_c(\boldsymbol{h}_i^l, \boldsymbol{h}_j^l)) \quad (19)$$

where $\text{Linear} : \mathbb{R}^d \to \mathbb{R}^{2d}$ ensures scale-awareness. Project $\boldsymbol{h}_i^x$, $\boldsymbol{h}_j^x$ and $\boldsymbol{h}_{ij}^e$ to $\mathcal{T}_0 \mathbb{B}_c^n$ via $\log_0^c(\cdot)$, concatenate, and modulate to update the message:

$$\boldsymbol{m}_i^{l+1} = \sum_{j \in \mathcal{N}(i)} \underbrace{\text{AdaLN}\left(\mathcal{W}_e\left[\log_0^c(\boldsymbol{h}_i^l), \log_0^c(\boldsymbol{h}_j^l), \log_0^c(\boldsymbol{h}_{ij}^l)\right], \boldsymbol{\gamma}_{ij}^l, \boldsymbol{\beta}_{ij}^l\right)}_{\boldsymbol{m}_{ij}^{l+1}}, \quad (20)$$

where $\mathcal{W}_e \in \mathbb{R}^{3n \times n \times d}$ is a Euclidean linear layer. We can get a new edge representation by pulling the message back up to the Poincaré manifold.

$$\boldsymbol{h}_{ij}^{l+1} = \exp_0^c\left(\boldsymbol{m}_{ij}^{l+1}\right) \quad (21)$$

**Hyperbolic Node Aggregation.** For node $i$, we have aggregated messages $\boldsymbol{m}_i^{l+1}$ from neighbors $\mathcal{N}(i)$ in Eq. 20. Similarly, we compute adaptive shift and scale parameters $\boldsymbol{\eta}_i^{l+1}, \boldsymbol{\zeta}_i^{l+1}$ from the message and update node features:

$$\boldsymbol{h}_i^{l+1} = \exp_0^c \left( \log_0^c(\boldsymbol{h}_i^l) + \text{AdaLN}\left( \mathcal{W}_x \left[ \log_0^c(\boldsymbol{h}_i^l), \log_0^c(\boldsymbol{m}_i^{l+1}) \right], \boldsymbol{\eta}_i^{l+1}, \boldsymbol{\zeta}_i^{l+1} \right) \right) \qquad (22)$$

with $\mathcal{W}_e \in \mathbb{R}^{2n \times n \times d}$ as Euclidean linear layer.

### E.6.3 POINCARÉ TRANSFORMERS

We present a hyperbolic transformer architecture that extends self-attention mechanisms to the Poincaré ball model while preserving geometric consistency. Given the hidden feature $\boldsymbol{H} = \{\boldsymbol{h}_i \in \mathbb{B}_c^n\}_{i=1}^N$, the transformer first injects positional information via Möbius addition of learnable hyperbolic embeddings. The sequence then passes through $L$ identical layers, which containing Poincaré Multi-Head Attention and Poincaré Feed-Forward Network. Both components employ residual connections with Möbius addition and Poincaré LayerNorm.

**Poincaré Attention Mechanisms.** The attention mechanism generalizes scaled dot-product attention to hyperbolic space through three stages:

**Projection & Splitting.** Inputs are projected into queries $\boldsymbol{Q}$, keys $\boldsymbol{K}$, and values $\boldsymbol{V}$ via Poincaré FC layers (Eq. 15). As described in Shimizu et al. (2020), we use $\beta$-splitting to split each tensor into $h$ heads:

$$\boldsymbol{Q}_i, \boldsymbol{K}_i, \boldsymbol{V}_i = \text{Split}_c(\boldsymbol{Q}), \text{Split}_c(\boldsymbol{K}), \text{Split}_c(\boldsymbol{V}), \qquad (23)$$

where $\text{Split}_c$ scales dimensions via beta functions.

**Geodesic Attention Score.** Attention weights are computed using hyperbolic distance to maintain the hyperbolic geometry structure, which can be formulated as:

$$\alpha_{ij} = \text{softmax}(\tau d_c(\boldsymbol{Q}_i, \boldsymbol{K}_i) - \gamma), \qquad (24)$$

where $\tau$ is an inverse temperature and $\gamma$ is a bias parameter, which was proposed by Gulcehre et al. (2019). After applying a similarity function and obtaining each weight, the values are aggregated by Möbius gyromidpoint:

$$\boldsymbol{Z}_i = \sum_{j=1}^T [\boldsymbol{V}_j, \alpha_{ij}]_c := \frac{1}{2} \otimes_c \left( \frac{\sum_j \alpha_{ij} \lambda_c^{V_j} \boldsymbol{V}_j}{\sum_j |\alpha_{ij}| (\lambda_c^{V_j} - 1)} \right), \qquad (25)$$

the symbol $\frac{1}{2} \otimes_c \underline{\boldsymbol{x}} = \frac{\underline{\boldsymbol{x}}}{1 + \sqrt{1 - c\|\underline{\boldsymbol{x}}\|^2}}$, which is described in Ungar (2008).

**Multi-Head Composition.** Head outputs $\{\boldsymbol{Z}_i\}_{i=1}^h$ are combined via $\beta$-concatenation and out projection:

$$\boldsymbol{O} = \text{Concat}_c(\boldsymbol{Z}_1, ..., \boldsymbol{Z}_h) \oplus_c \mathcal{W}_O, \qquad (26)$$

where $\mathcal{W}_O$ is a Poincaré linear projection and $\text{Concat}_c$ inversely scales beta ratios from splitting.

**Time-aware Transformer Layers.** To integrate temporal dynamics into hyperbolic diffusion processes, we extend the Poincaré transformer with adaptive time-conditioned modulation. Each layer injects timestep embeddings $\boldsymbol{t}_{emb} \in \mathbb{R}$ through Euclidean affine transformations of normalized features. Following Euclidean Diffusion Transformer (Peebles & Xie, 2023), we regress time-conditioned gate $\boldsymbol{\alpha}_{ij}^l$, shift $\boldsymbol{\beta}_{ij}^l$, and scale $\boldsymbol{\gamma}_{ij}^l$ parameters from $\boldsymbol{t}_{emb}$:

$$\boldsymbol{\alpha}_{attn}^l, \boldsymbol{\beta}_{attn}^l, \boldsymbol{\gamma}_{attn}^l, \boldsymbol{\alpha}_{ffn}^l, \boldsymbol{\beta}_{ffn}^l, \boldsymbol{\gamma}_{ffn}^l, = \text{Linear}(\boldsymbol{t}_{emb}) \qquad (27)$$

where $\text{Linear} : \mathbb{R}^{n \times d} \to \mathbb{R}^{6n \times d}$ is Euclidean linear layer. These parameters drive an AdaLN modulation mechanism and modulated feature representation $\boldsymbol{h}_{new}$ is computed as:

$$\boldsymbol{h}_{new} = \boldsymbol{h} \oplus_c \left[ \boldsymbol{\alpha}_{ij}^l \cdot \text{PoincareAttention}(\exp_0^c(\text{AdaLN}(\log_0^c(\boldsymbol{h}), \boldsymbol{\beta}_{ij}^l, \boldsymbol{\gamma}_{ij}^l))) \right], \qquad (28)$$

where $\text{AdaLN}(\cdot)$ is the similar function in Diffusion Transformer (Peebles & Xie, 2023). Subsequently, $\boldsymbol{h}_{new}$ is processed similarly in the Poincaré Feed-Forward Networks, forming a complete Time-aware Transformer Layer, and we use $L$ Time-aware Transformer Layers to construct Poincaré Diffusion Transformer used in Section 3.3.

### E.7 HYPERBOLIC AUTOENCODER FOR GRAPH

#### E.7.1 HYPERBOLIC GRAPH AUTOENCODER (HGAE)

Graph autoencoders are unsupervised learning methods designed to learn low-dimensional representations of graphs. In hyperbolic space, graph autoencoders can more effectively capture the hierarchical structure of graphs. Our Hyperbolic Graph Autoencoder (HGAE) consists of a hyperbolic encoder and a hyperbolic decoder. The encoder maps graph nodes to points in hyperbolic space, while the decoder reconstructs the adjacency matrix of the graph.

**Graph Encoder.** The encoder first incorporates extra structural and spectral graph features—eigenvectors and eigenvalues of the graph Laplacian—to enrich global topological awareness. Such features boost the expressive capacity of graph neural networks and are helpful for training a powerful autoencoder structure (Beaini et al., 2021; Vignac et al., 2023). Since these features are not exposed to the graph generation process, we could theoretically construct any features that can increase the expressiveness of graphs. These features undergo initial Euclidean preprocessing via multilayer perceptrons (MLPs), ensuring effective fusion of multidimensional attributes. The result is mapped onto the Poincaré ball with an origin-centred exponential map:

$$\mathbf{X}_h = \exp_0\big(\mathrm{MLP}([\mathbf{X}, \mathbf{S}])\big), \qquad \mathbf{E}_h = \exp_0\big(\mathrm{MLP}(\mathbf{E})\big) \tag{29}$$

Subsequent hyperbolic graph neural network (GNN) layers aggregate localized structural patterns, while $L$ hyperbolic Transformer layers propagate information globally, yielding final hyperbolic embeddings $\mathbf{z}_i = \mathbf{h}_i^L \in \mathbf{H}^D$ that preserve multiscale structural semantics.

$$\mathbf{h}_i^{l+1} = \mathrm{Attention}_\theta(\mathbf{h}_i^l), \qquad \mathbf{h}^0 = \mathrm{GNN}_\theta(\mathbf{X}_h, \mathbf{E}_h) \tag{30}$$

**Probabilistic Graph Decoder.** The decoder architecture operates under the conditional independence assumption, where node attributes depend solely on their respective embeddings, while edge connectivity and features are conditionally dependent on node pairs. This factorization yields the joint reconstruction probability:

$$p_\phi(\mathbf{X}, \mathbf{E} \mid \mathbf{H}) = \prod_{i=1}^n p_\phi(\mathbf{x}_i \mid \mathbf{h}_i) \prod_{j=1}^n \prod_{k=1}^n p_\phi(\mathbf{e}_{jk} \mid \mathbf{h}_j, \mathbf{h}_k) \tag{31}$$

We first feed the latent repsentations $\mathbf{z}_i$ into a 2 layer Poincaré self attention model, to allow full interaction between nodes, and obtain node-level representations $\mathbf{h}_i$. For attribute reconstruction, we first project the hyperbolic feature on to $\mathcal{T}_0\mathbb{B}_c^n$, and then apply a Euclidean MLP-based prediction head followed by Softmax distribution.

$$p_\phi(\mathbf{x}_i \mid \mathbf{h}_i) = \mathrm{SoftmaxMLP}_\phi\left(\log_0^c(\boldsymbol{h}_i)\right) \tag{32}$$

Edge probabilities are derived from a comprehensive set of hyperbolic geometric features:

$$\mathbf{f}_{ij} = \left[\log_{\mathbf{0}}^c(\mathbf{h}_i), \log_{\mathbf{0}}^c(\mathbf{h}_j), \log_{\mathbf{h}_i}^c(\mathbf{h}_j), d_c(\mathbf{h}_i, \mathbf{h}_j), \cos\theta_{ij}\right] \tag{33}$$

where the log-map vectors represents the directional relationship in the tangent space, $d_c\left(\boldsymbol{h}_i, \boldsymbol{h}_j\right)$ computes the explicit hyperbolic distance encoding hierarchical relationships, and $\theta_{ij} = \angle_{\mathbb{H}}(0, \boldsymbol{h}_i, \boldsymbol{h}_j)$ captures relative angular positions The resulting $\mathcal{R}^{3D+2}$ feature vector is processed through an MLP with softmax output to predict:

$$p_\phi(\mathbf{e}_{ij} \mid \mathbf{h}_i, \mathbf{h}_j) = \mathrm{Softmax}\left(\mathrm{MLP}_\phi(\mathbf{f}_{ij})\right) \tag{34}$$

**Reconstruction Loss.** By jointly optimizing feature and topological reconstruction under hyperbolic geometric constraints, the model ensures latent embeddings retain both structural hierarchy and attribute-node relationships, outperforming conventional Euclidean counterparts in preserving complex graph semantics.

$$\mathcal{L}_{\mathrm{AE}} = \lambda_{\mathrm{node}}\, \mathcal{L}_{\mathrm{node}} + \lambda_{\mathrm{edge}}\, \mathcal{L}_{\mathrm{edge}} + \lambda_{\mathrm{reg}}\, \mathcal{L}_{\mathrm{reg}}, \tag{35}$$

$$\mathcal{L}_{\text{node}} = - \mathbb{E}_{(\mathbf{X},\mathbf{E}) \sim \mathcal{D}} \, \mathbb{E}_{\mathbf{h} \sim p_\theta(\mathbf{h}|\mathbf{X},\mathbf{E})} \sum_{i=1}^{n} \sum_{c=1}^{d_1} x_{ic} \, \log p_\phi(x_{ic} = 1 \mid \mathbf{h}_i). \tag{36}$$

$$\mathcal{L}_{\text{edge}} = - \mathbb{E}_{(\mathbf{X},\mathbf{E}) \sim \mathcal{D}} \, \mathbb{E}_{\mathbf{h} \sim p_\theta(\mathbf{h}|\mathbf{X},\mathbf{E})} \sum_{i=1}^{n} \sum_{j=1}^{n} \sum_{c=1}^{d_2} e_{ijc} \, \log p_\phi(e_{ijc} = 1 \mid \mathbf{h}_i, \mathbf{h}_j). \tag{37}$$

With edge reconstruction loss, node reconstruction loss, and regularization loss, and lambda being hyperparameters.

Training instability remains a fundamental challenge in hyperbolic neural networks, primarily due to the exponential growth of hyperbolic distance as points move away from the origin toward the Poincaré ball boundary. Empirical observations reveal that embeddings with small norms (*e.g.*, below 0.1) exhibit near-Euclidean behavior, while those approaching the boundary (norms > 0.8) frequently suffer from gradient instability during optimization. To mitigate this issue, we adopt an adaptive regularization approach inspired by recent advances in clipped hyperbolic representations. Unlike hard constraints that strictly enforce norm boundaries, we introduce a band loss composed of two hinge loss terms, which softly encourages hyperbolic embeddings $\mathbf{h}_i$ to remain within a stable intermediate region.

$$\mathcal{L}_{\text{band}} = \frac{1}{n} \sum_{i=1}^{n} \Big[ \big(\max\{0, \, \|\mathbf{z}_i\| - r_{\max}\}\big)^2 + \big(\max\{0, \, r_{\min} - \|\mathbf{z}_i\|\}\big)^2 \Big] \tag{38}$$

Where we set the inner margin $r_{min} = 0.2$, and the outer margin $r_{max} = 0.8$

### E.7.2 DEGREE-EDGE CONSISTENCY LOSS

To ensure structural coherence between node-level predictions and edge-level decoding, we introduce a degree–edge consistency loss. This loss does *not* attempt to match the global degree histogram; instead, it enforces local consistency for each node between (i) the expected degree implied by the edge decoder, (ii) the ground-truth degree, and (iii) the degree predicted by the node-level classifier.

For node $i$, let

$$d_i^{\text{edge}} = \sum_{j=1}^{N} p_\phi(e_{ij} \mid h_i, h_j)$$

denote the expected degree induced by the edge probabilities, and let

$$d_i^{\text{pred}} = \mathbb{E}[d_i \mid h_i] = \sum_{k=0}^{K} k \, p_\phi(\deg_i = k \mid h_i)$$

denote the expected degree predicted by the node-degree classifier. The full degree–edge consistency loss is then defined as:

$$\mathcal{L}_{\text{degree}} = \frac{1}{N} \sum_{i=1}^{N} \big( \|d_i^{\text{edge}} - d_i^{\text{gt}}\|^2 + \|d_i^{\text{edge}} - d_i^{\text{pred}}\|^2 \big).$$

This term encourages the local connectivity implied by the edge decoder to agree with both the true degree and the internal node-level prediction. Such consistency is particularly beneficial for abstract graph datasets, where degree patterns reflect structural roles and strongly correlate with motif and orbit statistics.

### E.7.3 HYPERBOLIC VARIATIONAL AUTOENCODER (HVAE)

Variational Auto-Encoders augment a classic auto-encoder with a probabilistic KL penalty, helping to reduce the high-variance in latent space and enable sampling. In hyperbolic space $\mathbb{H}^n$, the Euclidean Gaussian is no longer appropriate, so we adopt the Wrapped Normal distribution (Mathieu et al., 2019; Grattarola et al., 2019; Nagano et al., 2019), for both prior and posterior:

$$p(\mathbf{z}_i) = \mathcal{N}_{\mathbb{H}}(\mathbf{0}, \Sigma), \qquad q_\phi(\mathbf{z}_i \mid \mathbf{X}, E) = \mathcal{N}_{\mathbb{H}}\big(\boldsymbol{\mu}_\phi(\mathbf{x}), \, \boldsymbol{\Sigma}_\phi(\mathbf{x})\big). \tag{39}$$

where $\boldsymbol{\mu} = \text{PoincareLinear}(h_i) \in \mathbb{B}_c^n$, $\Sigma = \text{Softplus}\left(\text{Linear}\left(\log_0^c(h_i)\right)\right) \in \mathbb{R}_+^n$. We choose the prior standard deviation be isotropic and renormalized by the square root of the dimension of hyperbolic space, $\Sigma = \frac{1}{\sqrt{d}}\mathbf{I}$, to ensure sampled points don't get pushed to the boundary as dimensionality increases.

**Reconstruction Loss.** HVAE is trained by maximising the evidence lower bound, or equivalently minimising $\mathcal{L}_{\text{VAE}} = \underbrace{\mathbb{E}_{q_\phi}\left[-\log p_\theta(\mathbf{x}, \mathbf{e} \mid \mathbf{z})\right]}_{\text{Reconstruction}} + \beta \underbrace{\text{KL}\left(q_\phi(\mathbf{z} \mid \mathbf{x}, \mathbf{e}) \parallel p(\mathbf{z})\right)}_{\text{KL}}$, where the lambda term controls the reconstruction-generation trade-off (Higgins et al., 2017). Since closed-form solutions for these distributions are unavailable, we compute the KL term via Monte Carlo estimation.

We have also implemented a hyperbolic variational autoencoder (HVAE), using the wrapped normal distribution as the prior in latent space, and tested the reconstruction performance. However, we observe that sampling in high-dimensional hyperbolic space is highly unstable, and Monte Carlo estimation of the KL divergence often leads to numerical instability. As a result, the HVAE significantly underperforms in both reconstruction and generation tasks compared to HAE and HVQVAE.

### E.7.4 Hyperbolic Vector Quantized Variational AutoEncoder (HVQVAE)

We introduce a hyperbolic vector quantized variational autoencoder (HVQVAE) that discretizes latent representations while preserving hierarchical structure.

**Hyperbolic Codebook.** We propose a novel hyperbolic codebook that resides entirely within the Poincaré disk, ensuring all operations remain consistent with the hyperbolic geometry.

**Codebook Initialization.** The hyperbolic codebook $\mathcal{C} \in \mathbb{B}_c^n$ is initialized via manifold-aware random sampling or optimized through hyperbolic $k$-means clustering. This algorithm uses initial vectors are drawn from the Poincaré neural network, ensuring uniform coverage across hierarchical scales and minimizes the sum of squared geodesic distance, which is defined as Eq. 14. The algorithm terminates when the centroid movement after 10 iterations (Alg. 16). We choose random initialization which achieves a slightly better results, while K-means initialization method often offers a faster convergence,

**Quantization Method.** In the Poincaré VQVAE, quantization maps an input embedding $\boldsymbol{Z} \in \mathbb{B}_c^n$ to the nearest codebook vector using the hyperbolic geodesic distance. Formally, the quantized embedding $\mathbf{z}_q$ is obtained by selecting the codebook vector $\mathbf{c}_i$ that minimizes the geodesic distance.

**Loss Function.** The training objective combines three geometrically consistent components:

$$\mathcal{L}_{\text{HVQVAE}} = \lambda_1 \underbrace{\mathbb{E}_{p_\phi}\left[-\log p_\theta(\boldsymbol{x}, \boldsymbol{e} \mid \boldsymbol{z}_q)\right]}_{\text{Reconstruction}} + \lambda_2 \underbrace{\mathbb{E}_z[d_c^2(\text{sg}(\boldsymbol{z}_q), \boldsymbol{z})]}_{\text{Commitment}} + \lambda_3 \underbrace{\mathbb{E}_z[d_c^2(\boldsymbol{z}_q, \text{sg}(\boldsymbol{z}))]}_{\text{Consistency}}, \quad (40)$$

where reconstruction loss is same as VAE; commitment loss anchors latent codes to quantized vectors, and Codebook loss updates embeddings via straight-through gradient estimation $\text{sg}(\cdot)$. It is worth noting that since the hyperbolic codebook is in hyperbolic space and the model parameters are in Euclidean space, we use both a Riemannian optimizer and a traditional optimizer, and the same is true for the positional encoding in Transformer.

**Hyperbolic Exponential Moving Average.** To stabilize training and avoid mode collapse, we incorporate Hyperbolic Exponential Moving Average (EMA) for codebook updates.

**EMA Update.** Codebook vectors $\boldsymbol{c}_j \in \mathbb{B}_c^n$ update via Riemannian EMA to balance historical states and new features. We compute a new centroid $\boldsymbol{\mu}_j$ from assigned samples $\boldsymbol{z}_i$ with weights $w_{ij} = \mathbb{I}(\boldsymbol{z}_i \to \boldsymbol{c}_j)$, which is an indicator function that assigns a weight of 1 when the sample $\boldsymbol{z}_i$ is mapped to the codebook vector $\boldsymbol{c}_j$ and 0 otherwise:

$$\boldsymbol{\mu}_j = \frac{1}{2} \otimes_c \left( \frac{\sum_{i=1}^N w_{ij} \lambda_c^{\boldsymbol{z}_i} \boldsymbol{z}_i}{\sum_{i=1}^N |w_{ij}|(\lambda_c^{\boldsymbol{z}_i} - 1)} \right) \quad (41)$$

Similarly, we can update codebook vectors via Einstein midpoint (Ungar, hypernetworks++):

$$\boldsymbol{c}_j^{(t+1)} = \text{proj}_{\mathbb{B}_c^d} \left( \left[ \boldsymbol{c}_j^{(t)}, \beta \right]_c \oplus_c \left[ \boldsymbol{\mu}_j, 1 - \beta \right]_c \right) \tag{42}$$

where the $\text{proj}_{\mathbb{B}_c^d}$ is projection in Poincaré ball, $\beta$ is custom decay, and the weighted midpoint operation is defined as:

$$[\boldsymbol{x}, w]_c := \frac{w \lambda_c^{\boldsymbol{x}} \boldsymbol{x}}{1 + \sqrt{1 + cw^2(\lambda_c^{\boldsymbol{x}})^2 \|\boldsymbol{x}\|^2}} \tag{43}$$

**Dead Codes Revival.** To avoid the problem of inactive codebook entries, we implement an expiration mechanism. If a codebook entry is not frequently updated, it is replaced with a new codebook vector. This mechanism ensures that the codebook remains active and capable of efficiently representing the latent space, thereby preventing collapse during training.

This hyperbolic codebook design, paired with tailored loss function and updating method, enables the Poincaré VQVAE to effectively model complex hierarchical data while preserving the manifold's intrinsic properties, offering a robust foundation for discrete representation learning in hyperbolic spaces.

### E.8 AUTOREGSSIVE MODEL FOR LATENT DISTRIBUTION

Autoregressive modeling in the latent space has become a common approach, especially for generating high-fidelity images (Li et al., 2024; He et al., 2022; Razavi et al., 2019). Since we have already trained a VQ model, we perform autoregressive learning over the discrete latent tokens. During training, a causal mask is applied to ensure that the model only attends to past tokens, with a Poincare Transformer backbone model. During generation, we compute the geodesic distance between the predicted next token embedding and all entries in the codebook, and apply softmax sampling to select the next token.

$$z_i p(z_i | z_{<i}) = \text{SoftmaxQuantize}(\text{PoincareTransformer}(z_{<i})) \tag{44}$$

### E.9 FLOW MATCHING

Let $\mathbb{R}^d$ denote the data space where the data samples $x_t \in \mathbb{R}^d$. The variable $t \in [0, 1]$ represents the inference time, where $p_1(x_1)$ is the target distribution we aim to generate, and $p_0(x_0)$ is a base distribution that is easy to sample from.

Flow-based generative models, as introduced by Chen et al. (2018) and Lipman et al. (2023), define a time-varying vector field $v_t(x_t)$ that generates a probability path $p_t(x_t)$, transitioning between the base distribution $p_0(x_0)$ and the target distribution $p_1(x_1)$. By first sampling $x_0$ from $p_0(x_0)$, and then solving the ordinary differential equation: $\frac{d}{dt} x_t = v_t(x_t)$. Specifically, we define a loss function that matches the model's vector field $v_t(x)$ to the target vector field $u_t(x)$:

$$\mathbb{E}_{t \sim U(0,1), x_t \sim p(x_t)} \|v_t(x_t; \theta) - u_t(x_t)\|^2$$

However, in practice, we cannot explicitly obtain the probability path $p_t(x_t)$ or the vector field $v_t(x_t)$, and thus, we are unable to directly compute these quantities. Instead, a more commonly used approach in practice is Conditional Flow Matching (CFM), which provides a practical way to approximate these elements and facilitate the generation process. CFM defines the conditional vector field $v_{t|z}(x_t|z)$ to obtain the conditional probability path $p(z|x_t)$. By marginalizing this conditional vector field and conditional probability path, we obtain the marginal vector field and marginal probability path. Specifically, the marginal vector field $v_t(x_t)$ is defined as:

$$v_t(x_t) := \int v_{t|z}(x_t|z) p(z|x_t) dz,$$

where $p(z|x_t) = \frac{p_t(x_t|z)p(z)}{p(x_t)}$, which generates the marginal probability path $p_t(x_t) = \int p_t(x_t|z)p(z)dz$. Thus, the marginal vector field governs the generation of the marginal probability path.

The CFM loss is given by the following formula:

$$\mathbb{E}_{t \sim U(0,1), z \sim p(z), x_t \sim p(x_t|z)} \left[ \left\| v_\theta(x_t, t) - v_{t|z}(x_t|z) \right\|^2 \right].$$

It can be shown that this loss is equivalent to the FM loss (Lipman et al., 2023).

Chen & Lipman (2024) extends this principle to Riemannian manifolds by ensuring that all vector fields lie in the tangent space $\mathcal{T}_{x_t} \mathcal{M}$, and distances are measured using the Riemannian metric $\mathfrak{g}$. The loss becomes:

$$\mathbb{E}_{t \sim U(0,1), x_0 \sim p(x_0), x_1 \sim p(x_1), x_t \sim p_t(x_0, x_1)} \| v_\theta(x_t, t) - u_t(x_t|x_1, x_0) \|^2_{\mathfrak{g}},$$

### E.10 TRAINING DETAILS

We primarily selected the hyperbolic VQVAE as the base autoencoder model. Subsequently, we conducted two-stage training for the autoregressive model and the flow matching model to enhance the generation performance. Similar to the StableDiffusion (Rombach et al., 2022), we chose to combine the quantizer and the decoder, which applies on the latent representation. It is worth noting that for our model, we employed the optimizer AdamW (Loshchilov & Hutter, 2017) for the Euclidean parameters, and the Riemannian optimizer RiemannianAdam (Kochurov et al., 2020) for all Riemannian parameters, which include the codebook and the positional embedding in the transformer. Most of the other operations are carried out in the Euclidean space, and they are mapped to different spaces through the exponential map and the log map.

We provide the number of layers and latent space dimensions used in our models in Table 6. All models are trained on a single A6000 GPU. On the QM9 dataset, the vector-quantized autoencoder is trained for approximately 6 hours, followed by 12 hours of flow training. For the COMM20 and Ego-Small datasets, the autoencoder training takes about 2 hours, and flow training takes around 3 hours. Training details are provided in Table 7. The loss weights are set as follows: the coefficients for node and edge reconstruction are both set to 5.0, the L2 regularization loss coefficient is set to 1e-4, the commitment loss coefficient is set to 0.25, and the VQ loss coefficient is set to 1.0.

Table 6: Model Parameters across datasets.

| Dataset | Comm-Small/Ego-Small | QM9 |
|---|---|---|
| Hyperbolic Channels | 64 | 128 |
| Codebook Size | 32 | 512 |
| Encoder GNN Layers | 2 | 2 |
| Encoder Transformer Layers | 8 | 8 |
| Encoder Dropout | 0.1 | 0.1 |
| Decoder Transformer Layers | 2 | 2 |
| Decoder Dropout | 0.1 | 0.1 |
| Decoder Layers | 2 | 2 |
| Flow DiT Transformer Layers | 8 | 8 |

**Hyperparameter tuning.** Most hyperparameters in our implementation are fixed across all datasets; we do not perform dataset-specific grid search or tuning. Specifically:

- **Loss weights:** all coefficients (reconstruction loss, degree loss, VQ commitment/consistency loss, regularization loss) are set to constant values for different dataset, chosen to balance magnitudes at initialization and ensure stable optimization.
- **Codebook size:** selected via codebook perplexity monitoring during VQ-VAE training, to avoid codeword collapse or extreme fragmentation, which is a standard practice in VQ-based methods (Van Den Oord et al., 2017).
- **Curvature of hyperbolic latent space:** fixed to $c = 0.1$. We do not tune curvature per dataset, because a constant curvature offers stable geometry and avoids potential gradient instability or embedding degeneration.

Empirically, this fixed-hyperparameter design yields robust performance across diverse datasets (Comm20, Ego-Small, QM9) without per-dataset hyperparameter search.

Table 7: Optimizer settings across geometric parameter spaces and training stages.

| Parameter Space | Parameter | Autoencoder | Flow |
|---|---|---|---|
| Poincaré | Optimizer | Riemannian Adam | Riemannian Adam |
| | Learning Rate | 0.005 | 0.005 |
| | Weight Decay | 0 | 0 |
| | Betas | (0.9, 0.999) | (0.9, 0.999) |
| | Grad Clipping | 1.0 | 1.0 |
| Euclidean | Optimizer | AdamW | AdamW |
| | Learning Rate | 0.0005 | 0.0005 |
| | Weight Decay | 1e−12 | 1e−12 |
| | Betas | (0.9, 0.999) | (0.9, 0.999) |
| | Grad Clipping | 1.0 | 1.0 |

# F  EXTRA EXPERIMENTS

## F.1  VARIATIONAL AUTOENCODER

As shown in Table 8, HVQVAE consistently outperforms both HAE and HVAE across all reconstruction metrics. While HVAE suffers from instability introduced by KL regularization on curved manifolds, HVQVAE achieves both lower distributional discrepancies (e.g., degree and clustering MMD) and higher edge reconstruction accuracy. Notably, on Community-small, HVQVAE reduces degree error by over 95% compared to HVAE, and improves edge accuracy to 99.39%. These results confirm the benefit of discrete latent representations in stabilizing learning and preserving topological structures in hyperbolic space.

Table 8: Reconstruction performance of our baseline models HAE, HVAE, and HVQVAE on abstract graphs (Community-small, Ego-small).

| Model | Community-small | | | | Ego-small | | | |
|---|---|---|---|---|---|---|---|---|
| | Deg. ↓ | Clus. ↓ | Orb. ↓ | Edge Acc. ↑ | Deg. ↓ | Clus. ↓ | Orb. ↓ | Edge Acc. ↑ |
| HAE | 0.0008 | 0.0310 | 0.0007 | 99.10 | 0.0019 | 0.0250 | 0.0048 | 92.40 |
| HVAE | 0.0175 | 0.0910 | 0.2347 | 97.36 | 0.0050 | 0.0320 | 0.0390 | 89.82 |
| HVQVAE | **0.0004** | **0.0208** | **0.0005** | **99.39** | **0.0002** | **0.0194** | **0.0018** | **93.20** |

While variational autoencoders (VAEs) are a natural choice for probabilistic modeling, we found that the KL divergence term becomes numerically unstable when optimized in hyperbolic space. As shown in Fig. 4, the KL divergence fluctuates dramatically during training. This instability stems from the mismatch between the prior and posterior distributions in non-Euclidean geometry, which lacks a tractable and well-behaved KL formulation. As a result, the VAE often fails to converge to a coherent latent structure.

Moreover, despite moderate improvements in reconstruction accuracy, the edge prediction performance of HVAE remains significantly inferior to our proposed HVQVAE. These observations motivate our choice to adopt a discrete latent representation, which provides greater numerical stability and structural expressiveness.

In future work, we plan to explore alternative variational formulations and prior structures that may stabilize training and fully leverage the geometric inductive bias of hyperbolic space.

## F.2  ADDITIONAL EXPERIMENT: ZINC250K

To further evaluate the scalability and generalization capability of our proposed framework on large-scale sparse graphs, we extended our experiments to the **ZINC250k** dataset. ZINC250k consists of

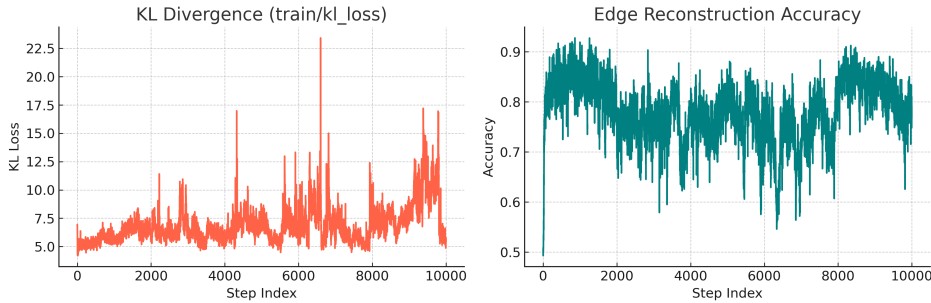

Figure 4: Training dynamics of the hyperbolic VAE (HVAE). Left: KL divergence exhibits large oscillations, indicating numerical instability. Right: Edge reconstruction accuracy improves slowly and remains suboptimal. These results highlight the limitations of using continuous probabilistic models in curved latent spaces.

approximately 250,000 drug-like molecular graphs, representing a significantly larger and structurally more complex domain compared to QM9 and Community-Small. This experiment serves to verify the model's performance in modeling realistic, sparse graph structures at scale.

Table 9: Generation performance on the ZINC250k dataset. We report the mean and standard deviation for GGBall over 3 independent runs. ↑ indicates higher is better. The best results are highlighted in **bold**.

| Model | Validity (%) ↑ | Uniqueness (%) ↑ | Novelty (%) ↑ | V.U.N ↑ |
|---|---|---|---|---|
| GraphAF | 68.00 | 99.10 | **100.00** | 67.38 |
| MoFlow | 81.76 | 99.99 | **100.00** | 81.75 |
| GraphDF | **89.03** | 99.20 | **100.00** | **88.32** |
| EDP-GNN | 82.97 | **100.00** | **100.00** | 82.97 |
| **GGBall (Ours)** | $83.50 \pm 0.98$ | $\mathbf{100.00} \pm 0.00$ | $98.88 \pm 1.01$ | $83.48 \pm 1.58$ |

**Discussion.** The quantitative results on ZINC250k are summarized in Table 9. GGBall achieves robust performance with a **Validity** of 83.50% and a **V.U.N.** score of 83.48, outperforming flow-based baselines such as GraphAF (67.38) and MoFlow (81.75), and performing comparably to the energy-based EDP-GNN (82.97). While GraphDF achieves a higher validity rate, GGBall maintains **100% Uniqueness** with high Novelty, indicating that our model can generate diverse and chemically valid structures without collapsing into repetitive patterns. These results confirm that the proposed hyperbolic framework generalizes effectively to large-scale real-world datasets.

### F.3 ADDITIONAL EXPERIMENT: TREE-STRUCTURED GRAPHS

Tree graphs constitute a prototypical example of hierarchical structure and are well aligned with the inductive bias of hyperbolic geometry. To assess how well our latent hyperbolic model captures such structure, we trained HVQVAE on the synthetic Tree dataset (Bergmeister et al., 2023) following the same protocol as in the main experiments.

**Results.** The hyperbolic autoencoder reconstructs trees with high accuracy across all structure-sensitive metrics.

| Model | Degree-MMD ↓ | Clustering-MMD ↓ | Orbit-MMD ↓ |
|---|---|---|---|
| HVQVAE (ours) | $\mathbf{7.9 \times 10^{-4}}$ | $\mathbf{2.6 \times 10^{-2}}$ | $\mathbf{1.0 \times 10^{-3}}$ |

**Discussion.** These results indicate that the hyperbolic latent space provides the correct geometric bias for modeling hierarchical connectivity. The extremely low Degree- and Orbit-MMD values

show that HVQVAE recovers the branching patterns and local motif statistics of trees with near-perfect accuracy. This supports our claim that hyperbolic latent structure is especially well suited for hierarchical graphs.

### F.4 ADDITIONAL EXPERIMENT: EUCLIDEAN ABLATION

We performed an ablation by replacing our full HVQVAE + hyperbolic-latent framework with a fully Euclidean counterpart (i.e. Euclidean GCN + Euclidean VQ-VAE + Euclidean Transformer). This isolates the effect of hyperbolic geometry while keeping network architecture, codebook, and training procedure unchanged. The results (on Community-small and Ego-Small) are summarized below:

Table 10: Reconstruction Performance: Hyperbolic vs. Euclidean Parametrization

| Model | Comm20 | | | Ego-Small | | |
|---|---|---|---|---|---|---|
| | Deg. ↓ | Clus. ↓ | Orb. ↓ | Deg. ↓ | Clus. ↓ | Orb. ↓ |
| Hyperbolic AE | 0.0008 | 0.0310 | 0.0007 | 0.0019 | 0.025 | 0.0048 |
| Hyperbolic VQVAE | **0.0004** | **0.0208** | **0.0005** | **0.0002** | **0.0194** | **0.0018** |
| Euclidean VQVAE | 0.0457 | 0.1344 | 0.7555 | 0.006 | 0.0815 | 0.0205 |

On Comm20, Euclidean VQ-VAE yields a larger Orbit-MMD (0.76) and lower structural fidelity, whereas our hyperbolic HVQVAE obtains near-zero Orbit-MMD and extremely low degree/clustering discrepancy. This dramatic gap suggests that the gains are not due to network architecture or codebook quantization per se, but indeed arise from the inductive bias of hyperbolic geometry.

When replacing the Poincaré parametrization with Euclidean space, the model largely loses the ability to compactly and faithfully represent hierarchical or community-structured graphs, confirming that the hyperbolic latent space is the main driver of performance improvements.

### F.5 STABILITY ANALYSIS VIA MEAN/STD REPORTING

To assess the robustness of our method beyond single-run performance, we report the mean and standard deviation of reconstruction metrics across 3 random seeds. As shown in Table 11, the hyperbolic VQVAE consistently achieves low reconstruction error and high edge accuracy with only minor fluctuations across runs, indicating that the model behaves stably under different initializations.

On the abstract graph datasets (Ego-Small and Community-Small), degree, clustering, and orbit MMD values remain tightly concentrated, and reconstruction accuracy remains above 98% across seeds. For the molecular datasets (QM9 and ZINC250k), validity, uniqueness, and novelty also show narrow confidence ranges, demonstrating stable behavior in both reconstruction and generation.

Overall, the small variance observed across all datasets confirms that the performance gains reported in the main text are consistent and reproducible, rather than the result of a favorable random seed or training instability.

Table 11: Reconstruction and generation stability across multiple seeds (Mean ± Std).

| Dataset | Deg. ↓ | Clus. ↓ | Orb. ↓ | Edge Acc. |
|---|---|---|---|---|
| **Ego-Small** | $0.0012 \pm 0.0007$ | $0.0237 \pm 0.0006$ | $0.0017 \pm 0.0003$ | $99.03\pm0.33\%$ |
| **Community-Small** | $0.0008 \pm 0.0012$ | $0.0215 \pm 0.0012$ | $0.0009 \pm 0.0010$ | $98.49\pm1.12\%$ |
| | **Validity ↑** | **Unique ↑** | **Novelty ↑** | **Edge Acc.** |
| **QM9** | $97.42 \pm 1.00$ | $98.96 \pm 0.46$ | $93.55 \pm 2.70$ | $99.21\pm0.30\%$ |
| **ZINC250k** | $90.01 \pm 2.36$ | $100.00 \pm 0.00$ | $99.47 \pm 0.32$ | $99.39\pm0.26\%$ |

## G  EFFICIENCY ANALYSIS

We conduct a targeted efficiency analysis to quantify the practical benefits of our architectural design. Unlike dual-space diffusion models such as HGDM, which require iterative denoising over both nodes and edges with separate networks, GGBall employs a unified hyperbolic latent space for nodes, from which the entire graph is decoded. This design avoids the computational overhead and complexity of explicit, iterative edge modeling. We compare inference performance on community graphs and a batch size of 128, with results summarized in Table 12.

Table 12: Inference efficiency comparison on Community-small (N=18) with a batch size of 128. Models marked with * are the primary configurations used in the main generation benchmarks. Lower is better for memory and time.

| Model (Inference Steps) | Params (M) | GPU Memory (MB) $\downarrow$ | Time/Batch (s) $\downarrow$ |
|---|---|---|---|
| HGDM (100 steps) | 0.18 | 342.3 | 13.49 |
| HGDM (10 steps) | 0.18 | 342.3 | 9.03 |
| GGBall (100 steps)* | 1.64 | **151.8** | 12.45 |
| GGBall (10 steps) | 1.64 | **151.8** | 1.40 |
| GGBall-Oneshot (1 steps) | 0.64 | **110.8** | **0.04** |

Table 12 highlights GGBall's significant advantages in two key areas. First, it is remarkably memory-efficient. Despite having approximately 9 times more parameters in its diffusion configuration, our model uses less than half the GPU memory of HGDM (151.8 MB vs. 342.3 MB). This stems directly from our unified architecture, which eliminates the need for a separate, memory-intensive edge-denoising network and its associated intermediate activations.

Second, GGBall offers superior inference speed and flexibility. Our benchmark model (GGBall*) is already faster than a comparable HGDM configuration with the same number of steps. More importantly, our framework uniquely supports a one-shot generation mode (GGBall-Oneshot), capable of decoding an entire batch of graphs in a mere 0.04 seconds. This near-instantaneous generation capability stands in stark contrast to the iterative nature of diffusion models, unlocking possibilities for applications requiring real-time or large-scale graph deployment.

## H  ADDITIONAL DISCUSSION: HIGHER-ORDER STRUCTURAL DEPENDENCIES

Although our decoder factorizes the likelihood into node-wise and pairwise edge terms for computational efficiency, this design does *not* prevent the model from capturing rich higher-order dependencies. The latent hyperbolic Transformer aggregates global information through attention layers, producing latent embeddings that already encode hierarchical, long-range, and motif-level patterns prior to decoding.

Furthermore, the framework naturally supports extensions to *explicit* higher-order decoding. For example, one may incorporate a triangle-consistency objective by matching the expected triangle count of each node:

$$t_i^{\mathrm{pred}} = \sum_{j,k} p_{ij}\, p_{ik}\, p_{jk} \quad \text{to} \quad t_i^{\mathrm{gt}},$$

where $p_{ij}$ denotes the predicted edge probability. Alternatively, a triadic potential module can be added:

$$s_{ijk}^{(\triangle)} = \mathrm{MLP}\left([\log(x_i)\| \log(x_j)\| \log(x_k)]\right),$$

whose outputs can be integrated into the pairwise logits $\tilde{s}_{ij}$ to provide motif-aware corrections.

Importantly, these extensions require *no change* to the encoder, hyperbolic latent geometry, or the VQ representation. The architecture is therefore fully compatible with $k$-wise hyperbolic potentials and higher-order structural modeling. We view explicit motif-aware hyperbolic decoders as a promising direction for future work, and our current design is intentionally structured to accommodate such extensions seamlessly.

## I  ADDITIONAL DISCUSSION: HEAD-TO-HEAD COMPARISON WITH HGDM

It is helpful to explicitly contrast our latent-space design with that of HGDM (Wen et al., 2024) to clarify where the novelty and trade-offs lie with other hyperbolic graph generation methods.

- HGDM performs diffusion in a hybrid latent space. Specifically, node features $\mathbf{X}_t$ evolve in a hyperbolic space $\mathbb{B}_c^d$, edge features $\mathbf{A}_t$ in Euclidean space. Both are separately updated during generation using two independent score networks:

$$dX_t = [f_{1,t}(X_t) - g_{1,t}^2 \nabla_{X_t} \log p_t(X_t, A_t)] \, dt + g_{1,t} \, d\bar{w}_1$$

$$dA_t = [f_{2,t}(A_t) - g_{2,t}^2 \nabla_{A_t} \log p_t(X_t, A_t)] \, dt + g_{2,t} \, d\bar{w}_2$$

- In contrast, GGBall maps the entire graph $G = (\mathbf{X}, \mathbf{A})$ into a unified, node-only representation in hyperbolic latent space. Formally, we embed the whole graph into node embeddings $\mathbf{Z} \in (\mathbb{B}_c^d)^N$, where $N$ is the number of nodes and $\mathbb{B}_c^d$ is the Poincaré ball.
  - This representation is both compact and information-rich: edge connectivity is instead implicitly encoded in node embeddings.
  - During generation, we only sample and evolve these node embeddings inside $\mathbb{B}_c^d$, and a decoder reconstructs the full graph from the node-only representation. This makes GGBall a truly hyperbolic latent-space generation model.

## J  SAMPLING STRATEGY

In large-scale graphs, naïvely decoding all pairwise edges leads to $\mathcal{O}(n^2)$ complexity, and without constraints, the number of sampled edges can grow exponentially with the number of nodes. To address this challenge, we introduce a principled strategy to control edge sparsity by aligning the predicted number of edges with that of real graphs.

**Temperature-Controlled Sampling.**  At generation time, we avoid fixed-threshold binarization or uncalibrated Multinomial sampling, which can lead to dense or overly sparse graphs. Instead, we introduce a learnable temperature parameter $\tau$ to scale the edge logits:

$$p_{ij} = \sigma(s_{ij}/\tau), \tag{45}$$

where $s_{ij}$ is the raw logits for edge $(i, j)$. The temperature $\tau$ is adjusted gradually on the validation set, such that the expected number of sampled edges, $\sum_{i \neq j} p_{ij}$, approximates the desired edge count. This provides a dynamic mechanism to modulate edge density as a function of graph size and latent structure.

Together, this strategy ensures that the number of generated edges respects the dataset feature, rather than quadratically with the number of nodes, preserving both scalability and fidelity to real-world graph sparsity .

## K  STATISTICAL METRICS

### K.1  GENERIC GRAPH GENERATION AND RECONSTRUCTION

To assess the structural fidelity of generated graphs, we compare the distributions of key graph statistics with those from the reference dataset, including:

- **Degree distribution**, reflecting node connectivity,
- **Clustering coefficient**, measuring local triangle density,
- **Orbit counts**, capturing subgraph motif frequencies.

We compute the **Maximum Mean Discrepancy (MMD)** between the empirical distributions of these statistics for generated and ground truth graphs. The MMD is computed with a Gaussian kernel:

$$\text{MMD}^2 = \mathbb{E}[k(G, G')] + \mathbb{E}[k(H, H')] - 2\mathbb{E}[k(G, H)],$$

where $G, G'$ are generated graphs and $H, H'$ are reference graphs.

### K.2 MOLECULAR GRAPH GENERATION

We evaluate the quality of generated graphs using four standard metrics: **Validity**, **Uniqueness**, **Novelty**, and **Distributional Fidelity**.

- **Validity** measures the proportion of generated graphs that are structurally valid, i.e., they satisfy predefined syntactic or domain-specific constraints (*e.g.*, chemical valency rules in molecular graphs). Formally, if $n_s$ graphs are generated and $V$ of them are valid, then Validity $= |V|/n_s$.

- **Uniqueness** evaluates the diversity of valid samples by computing the fraction of non-isomorphic graphs among the valid set. If $C$ is the set of valid graphs and $\text{set}(C)$ the set of unique ones, then Uniqueness $= |\text{set}(C)|/|C|$.

- **Novelty** quantifies the generative model's ability to produce new, unseen structures. It is defined as the proportion of valid unique graphs that are not present in the training dataset. Let $D_{\text{train}}$ denote the training graph set, then Novelty $= 1 - |\text{set}(C) \cap D_{\text{train}}|/|\text{set}(C)|$.

- **Distributional Fidelity** evaluates how closely the distribution of generated samples matches that of the real data. We use the *Fréchet ChemNet Distance (FCD)* (Preuer et al., 2018) as the metric, which computes the Fréchet distance between the feature distributions of generated and real molecules. Specifically, each molecule is encoded using the penultimate layer of a pretrained ChemNet model. The resulting embeddings are assumed to follow multivariate Gaussian distributions. Let $(m, C)$ and $(m_w, C_w)$ denote the mean and covariance of embeddings from generated and real molecules, respectively. The FCD is defined as:

$$d^2 = \|m - m_w\|_2^2 + \text{Tr}(C + C_w - 2(CC_w)^{1/2}),$$

which corresponds to the 2-Wasserstein distance. Lower FCD indicates that the generated samples are closer in distribution to the real data.

### K.3 EVALUATION PROTOCOLS

We follow the standard evaluation practices established in prior graph generation literatures (Vignac et al., 2023; Wen et al., 2024; Boget et al., 2023). For all datasets used in this work (COMM20, Ego-Small, and QM9), we adopt the official train/validation/test splits released by these benchmarks to ensure comparability and reproducibility.

At test time, we generate a batch of graphs equal in size to the test set and compute the corresponding MMD-based structural metrics (degree-, clustering-, and orbit-MMD) or chemical metrics (validity, uniqueness, novelty) depending on the dataset. All metrics are computed exactly following the implementations used in prior works to avoid discrepancies due to evaluation code.

**Dataset Splits.** For each abstract dataset, with the same seed from the original paper, we construct the train/validation/test partitions in three steps. First, we reserve $20\%$ of all graphs as the test set. From the remaining graphs, we take $80\%$ as the training set, and the remainder becomes the validation set. This results in approximate proportions of $64\%$ training, $16\%$ validation, and $20\%$ testing. For the QM9 molecular dataset, we assign 100k molecules to the training set, 10% of the full dataset to the test set, and use the remaining molecules as the validation split.

**Checkpoint Selection.** To guarantee full reproducibility and prevent test leakage, all model selection is performed using validation data:

- **VQ-VAE stage:** We select the checkpoint with the lowest validation reconstruction structural divergence, measured as the sum of MMD scores across degree, clustering, and orbit distributions, or the product of molecular validity, uniqueness, and novelty.

- **Flow Matching stage:** We select the checkpoint achieving the lowest validation generative structural divergence.

This two-stage evaluation protocol follows standard practice and ensures that test metrics reflect the generative model's true performance rather than hyperparameter search or checkpoint selection on the test split.

## L    LIMITATIONS & FUTURE WORK

While our current experiments focus on small to medium-scale graphs such as Community-Small, Ego-Small, and QM9, a key direction is to scale our method to much larger graph domains (e.g. social networks, citation graphs, molecular datasets with thousands of atoms). It is not yet clear whether the inductive biases of hyperbolic latent geometry will continue to offer advantages at scale.

Another exciting direction is to explore mixed-curvature latent spaces. Recent work in knowledge graphs and representation learning has shown that embedding in a single curvature space (pure hyperbolic or Euclidean) can be limiting when the data exhibits heterogeneous structural patterns.

For example, M2GNN embeds KGs in a product of different curvature spaces and shows performance gains in multi-relational reasoning tasks (Wang et al., 2021b). Also, more recent studies examine learning mixed geometry in latent models (e.g. Yusupov et al. (2025) discuss expressive capacity of mixed geometry for embeddings). Such a hybrid geometry paradigm may allow our model to adaptively allocate curvature to different substructures, providing both flexibility and structural fidelity.

Finally, we plan to validate our model in more application-oriented downstream tasks, such as conditional molecule design (e.g. generating molecules with target properties), knowledge graph augmentation, or graph-based program synthesis. Extending our hyperbolic latent framework to such tasks would enhance its utility and demonstrate applicability beyond benchmarking.

