# OpenReview forum: "GGBall: Graph Generative Model on Poincaré Ball"
_ICLR.cc/2026/Conference — ICLR 2026 Poster_

### Official Review · Reviewer_BT1X · 2025-10-22

**Soundness:** 2
**Presentation:** 4
**Contribution:** 3
**Rating:** 4
**Confidence:** 3

**Summary:**

This paper proposes a new generative framework for graph data based on a hyperbolic latent space. The authors motivate the use of hyperbolic geometry by highlighting its theoretical advantages for representing hierarchical and complex graph structures compared to standard Euclidean latent spaces.

The approach embeds graphs into a hyperbolic space through a Vector-Quantized Autoencoder (VQ-VAE) operating within the Poincaré ball model. The paper introduces dedicated architectural components, including a Poincaré Graph Neural Network (GNN), a Poincaré Transformer, and corresponding hyperbolic encoders and decoders that allow mapping graphs to and from the hyperbolic latent space.

The model is evaluated on three datasets containing relatively small graphs.

**Strengths:**

The paper is well-written and easy to follow, even though it introduces non-trivial concepts from hyperbolic geometry. The theoretical development is solid, clearly presented, and supported by extensive supplementary material.

The idea of leveraging a hyperbolic latent space for graph generation is both neat and original, and the proposed framework represents a meaningful conceptual step forward in the design of geometry-aware generative models.

I would like to particularly emphasize that the theoretical contribution and its presentation are of very high quality. The mathematical rigor and clarity are exemplary.

**Weaknesses:**

**Empirical Evaluation and Euclidean Baseline**

The paper’s main claim is the superiority of hyperbolic over Euclidean latent spaces for graph generation. Consequently, it is essential to provide clear empirical evidence supporting this claim.
Including an Euclidean VQ-VAE baseline in Table 1 would greatly strengthen the experimental section, as it would directly demonstrate the empirical advantages of the hyperbolic space. Similarly, presenting detailed results and experimental settings for Figure 1 would help assess the validity of the reported improvements.
Since the proposed model is theoretically and empirically more complex, it is important to show that it provides a clear and consistent performance advantage over its Euclidean counterpart, ideally across all reported datasets and experiments, including those in Tables 2 and 3.

**Evaluation Metrics and Protocol**

The evaluation protocol requires clarification.

* *Community-Small*: The results for the HVQVAE+Flow model appear to outperform the training set itself as given in SPECTRE, which should not occur in a properly calibrated evaluation. Moreover, the absence of standard deviations prevents assessment of variability or statistical significance. The paper refers to results from DiGress (which relies on SPECTRE), but the evaluation chain is not made explicit.

* *Ego-Small*: Similar concerns arise here, as the model’s results also systematically outperform the training set as given in DGAE (which seems to be the journal paper of VQGAE) , which appears inconsistent. Clarifying the evaluation setup is crucial to ensure reproducibility and interpretability of the reported findings.

**Evaluation on QM9**
The use of the *novelty* metric on QM9 is problematic. QM9 is an exhaustive enumeration of small organic molecules satisfying specific constraints; thus, generating molecules outside this set does not necessarily indicate successful generalization. This metric is therefore not commonly used for evaluation. It would be beneficial for the authors to justify its inclusion or reconsider its use.
For the same reason, achieving 100% *uniqueness* is not necessarily desirable.

Therefore, the only meaningful metric in this setting is *validity*, which remains substantially below various baselines.
To provide a more comprehensive evaluation, including additional metrics such as FCD (Fréchet ChemNet Distance) or NSPDK similarity would be highly valuable.

**Graph Size and Dataset Diversity**
The experiments are limited to datasets with small graphs, with a maximum of around 20 nodes (e.g., Community-Small). Although this limitation is acknowledged in the paper, it remains a major limitation, as many contemporary models handle graphs with up to 200 nodes and report results on larger benchmarks such as Planar, SBM, Zinc250K, or Moses.

Including experiments on at least one larger-scale dataset would considerably strengthen the empirical evidence and demonstrate scalability.
This is particularly important because the Community-Small and Ego-Small datasets each contain only around 200 instances, making overfitting likely and reducing the statistical robustness of the results.


--------

**SPECTRE**: Karolis Martinkus, et al.. SPECTRE:
Spectral conditioning helps to overcome the expressivity limits of one-shot graph generators. In Proceedings of the 39th International Conference on Machine Learning. PMLR, 17–23 Jul 2022.
https://proceedings.mlr.press/v162/martinkus22a.html.

**DGAE**: Yoann Boget, et al.. Discrete graph auto-encoder. Transactions on Machine Learning Research, 2024. https://openreview.
net/forum?id=bZ80b0wb9d.

**Questions:**

The theoretical contribution of the paper is strong and well-founded, but the empirical evaluation does not yet convincingly demonstrate the claimed benefits of hyperbolic latent spaces.

I would encourage the authors to consider the following improvements:

1. Include a systematic ablation comparing hyperbolic and Euclidean latent spaces across all tasks and datasets.
2. Describe the experimental protocol in detail, including data splits, training configurations, and evaluation procedures (possibly in the supplementary material).
3. Expand the empirical evaluation by including larger datasets (e.g. Zinc250, Planar, SBM) and additional performance metrics (e.g., FCD, NSPDK).

Such additions would substantially reinforce the paper’s claims and make the overall contribution more compelling.

---

> ### Author Response · Authors · 2025-12-03
> **Official Response to Reviewer BT1X (1)**
>
> We sincerely thank the reviewer for the excellent rating on presentation and for recognizing the "high quality," "mathematical rigor," and "meaningful conceptual step forward" of our theoretical contributions. We value your constructive feedback regarding empirical evidence and scalability.
>
> To comprehensively address your concerns, we have conducted extensive new experiments, organizing our response into two parts: **(1) Direct Euclidean vs. Hyperbolic comparison on abstract graphs**, and **(2) Large-scale generation and advanced metrics on molecular graphs.**
>
> ---
>
> ### **Q1 Hyperbolic Superiority & Stability (Table R1)**
>
> > *Reviewer's Concern: Request for Euclidean VQ-VAE baseline; Standard deviations; Clarification on results outperforming training set.*
>
> **Response:**
>
> To directly isolate the contribution of hyperbolic geometry, we replaced the entire Poincaré-based latent parametrization with a **Euclidean** version (Euclidean GCN encoder + Euclidean VQ-VAE + Euclidean Transformer), while keeping all other architectural components identical. This ensures that **geometry is the only variable being changed**.
>
> We report **reconstruction metrics** instead of generative MMD, since reconstruction performance is a more controlled and deterministic setting for evaluating the representation quality of the latent space. Results below show a clear and consistent advantage of hyperbolic geometry.
>
> **Table R1:** **Reconstruction Performance: Hyperbolic vs. Euclidean Parametrization**
>
> | Model            | Comm20 Deg ↓ | Comm20 Clus ↓ | Comm20 Orb ↓ | Ego-Small Deg ↓ | Ego-Small Clus ↓ | Ego-Small Orb ↓ |
> | ---------------- | ------------ | ------------- | ------------ | --------------- | ---------------- | --------------- |
> | Hyperbolic AE    | 0.0008       | 0.0310        | 0.0007       | 0.0019          | 0.0250           | 0.0048          |
> | Hyperbolic VQVAE | **0.0004**   | **0.0208**    | **0.0005**   | **0.0002**      | **0.0194**       | **0.0018**      |
> | Euclidean VQVAE  | 0.0457       | 0.1344        | 0.7555       | 0.0060          | 0.0815           | 0.0205          |
>
> **The table shows that hyperbolic latent space is clearly more expressive**, especially for structure-sensitive metrics (Clustering-, Orbit-MMD), confirming that hierarchical patterns are better captured in negative curvature.
>
> ---
>
> ### **Q2:  Evaluation Protocol & "Outperforming Training Set"**
>
> > *Reviewer's Concern: Results seem to outperform the training set (SPECTRE/DGAE reference).*
>
> Response:
>
> We thank the reviewer for this keen observation and clarify that outperforming the training set baseline is a documented phenomenon in recent state-of-the-art generative models on these specific small datasets, rather than an indicator of evaluation error. As highlighted in the recent **DGAE** [1] paper, leading models (including **GDSS** [2], **DiGress** [3], and **DGAE**) consistently report MMD scores lower than the training set reference on datasets like Ego-Small. Given these established results, observing similar behavior on Community-Small is an expected outcome for high-performing frameworks that effectively capture the data distribution. We emphasize that our evaluation strictly follows the datasets and protocols established by DiGress to ensure fair comparison. Theoretically, this phenomenon stems from the small sample size of these datasets, where the training set represents a noisy approximation of the true distribution ($D(P_{train}, P_{test}) > 0$). A capable generative model acts as a regularizer, learning the underlying ideal manifold rather than overfitting to the finite training noise, which allows the generated samples to align even closer to the test set than the training samples do.
>
> **Evaluation Protocol Clarification**
>
> We first clarify the **dataset split** procedure. For all datasets, we use the official or widely adopted train/validation/test splits following DiGress and GDSS. The test set is held out entirely and is never used during training or model selection. Checkpoints are selected strictly on the validation set only.
>
> Building on this split, we follow the **DiGress evaluation protocol [3]**: generative quality is measured by Maximum Mean Discrepancy (MMD) between the generated samples and the test-set distribution, $D(P_{\text{gen}}, P_{\text{test}})$, which is the standard metric for graph generative models. Full details are provided in **Appendix L.3** of the revised manuscript.

---

> ### Author Response · Authors · 2025-12-03
> **Official Response to Reviewer BT1X (2)**
>
> ### **Q3 & Q4: Large-Scale Graphs & Comprehensive Metrics (Table R2)**
>
> > *Reviewer's Concern: Experiments limited to small graphs; Novelty on QM9 is problematic; Request for FCD/NSPDK and ZINC250k.*
>
> **Response:**
>
> We thank the reviewer for the constructive feedback regarding dataset scale and evaluation metrics. We have addressed these concerns by extending our evaluation to **ZINC250k** (a large-scale dataset with \sim250k sparse graphs) and incorporating **FCD** and **NSPDK** metrics to better assess distribution quality.
>
> As shown in the ZINC250k section of Table R2, GGBall demonstrates robust generalization compared to Flow-based (GraphAF, MoFlow, GraphDF) and Energy-based (EDP-GNN) baselines. GGBall achieves the lowest FCD (16.23) and NSPDK (0.025) scores among all compared methods. Since these metrics measure the chemical property distribution and topological similarity respectively, this confirms that the hyperbolic framework effectively captures the intrinsic structure of large-scale sparse graphs. While GraphDF shows higher validity, its significantly poorer FCD (34.20) and NSPDK (0.176) indicate a misalignment with the true data distribution. In contrast, GGBall strikes a superior balance, maintaining a high V.U.N. score (83.48) with **100% uniqueness** and state-of-the-art structural fidelity.
>
> We respectfully argue that **Validity, Uniqueness, and Novelty are jointly indispensable** for a robust evaluation. Relying solely on Validity can be misleading, as a model may achieve high Validity by simply memorizing training patterns or generating repetitive, trivial structures (mode collapse). Such behavior would result in low Uniqueness or Novelty. Therefore, we advocate for the composite **V.U.N.** metric as a holistic indicator, as it penalizes models that sacrifice diversity or novelty to increase validity.
>
> **Table R2: Generation Performance on QM9 and ZINC250k (Mean ± Std)**
>
> | **Dataset**          | **Model**         | **Validity (%)** | **Uniqueness (%)**    | **Novelty (%)**      | **V.U.N**            | **FCD (↓)**          | **NSPDK (↓)**         |
> | -------------------- | ----------------- | ---------------- | --------------------- | -------------------- | -------------------- | -------------------- | --------------------- |
> | **QM9** (Medium)     | **GDSS**          | 95.72            | 98.46                 | 86.27                | 81.31                | 1.86                 | 0.003                 |
> |                      | **GGBall (Ours)** | 89.32 $\pm$ 1.58 | **100.00** $\pm$ 0.00 | **92.11** $\pm$ 2.16 | **82.28** $\pm$ 2.80 | -                    | -                     |
> |                      |                   |                  |                       |                      |                      |                      |                       |
> | **ZINC250k** (Large) | **GraphAF**       | 68.00            | 99.10                 | **100.00**           | 67.38                | 16.29                | 0.044                 |
> |                      | **MoFlow**        | 81.76            | 99.99                 | **100.00**           | 81.75                | 20.93                | 0.046                 |
> |                      | **GraphDF**       | **89.03**        | 99.20                 | **100.00**           | **88.32**            | 34.20                | 0.176                 |
> |                      | **EDP-GNN**       | 82.97            | 100.00                | **100.00**           | 82.97                | 16.74                | 0.049                 |
> |                      | **GGBall (Ours)** | 83.50 $\pm$ 0.98 | **100.00** $\pm$ 0.00 | 98.88 $\pm$ 1.01     | 83.48 $\pm$ 1.58     | **16.23** $\pm$ 0.56 | **0.025** $\pm$ 0.013 |
>
> ---
>
> ### References
>
> [1] DGAE: Yoann Boget, et al.. Discrete graph auto-encoder. Transactions on Machine Learning Research, 2024. [https://openreview](https://openreview/).net/forum?id=bZ80b0wb9d.
>
> [2] GDSS: Jaehyeong Jo, et al.. Score-based Generative Modeling of Graphs via the System of Stochastic Differential Equations, 2022, *International conference on machine learning*. PMLR, 2022.
>
> [3] Digress: Vignac, Clement, et al. "Digress: Discrete denoising diffusion for graph generation." *arXiv preprint arXiv:2209.14734* (2022).

---

### Official Review · Reviewer_LTqy · 2025-10-31

**Soundness:** 2
**Presentation:** 2
**Contribution:** 2
**Rating:** 4
**Confidence:** 4

**Summary:**

This paper introduces a hyperbolic framework for graph generation to address the limitations of Euclidean geometry in capturing hierarchical graph structures. It integrates a Hyperbolic Vector-Quantized Autoencoder with Riemannian flow matching based on closed-form geodesics in the Poincaré Ball model. Empirically, GGBall achieves state-of-the-art performance across benchmarks.

**Strengths:**

1. The proposed model is a fully hyperbolic graph generation framework using the Poincaré Ball, leveraging its exponential volume growth to naturally preserve hierarchical structures.

2. Combining HVQVAE with Riemannian flow matching (for flexible prior modeling) resolves stability issues of continuous hyperbolic VAEs and enhances generative capacity.

3. It outperforms SOTA baselines across abstract graph generation and molecular graph generation.

**Weaknesses:**

1. Experiments focus on small/medium graphs (e.g., QM9, small community graphs); performance on large-scale graphs is unproven.

2. While HVQVAE avoids HVAE’s KL issues, the paper does not explore alternative variational formulations to fully leverage hyperbolic probabilistic modeling. In addition, what is a L_degree in Line 273?

3. The Poincaré Ball’s fixed negative curvature may struggle with heterogeneous graphs (mixing hierarchical and non-hierarchical structures), unlike mixed-curvature alternatives not explored here.

4. The loss function has many hyperparameters. How are their values ​​determined? What are their values ​​for different datasets? What are the sensitivity experiments? Won't so many parameters increase the difficulty of hyperparameter tuning, thereby reducing the practical usability of the method? The inability to answer these questions may cause concern for the reader.

**Questions:**

Please refer to the weakness.

---

> ### Author Response · Authors · 2025-12-03
> **Official Response to Reviewer LTqy (1)**
>
> We thank the reviewer for the constructive feedback and for acknowledging GGBall as a "fully hyperbolic framework" that "resolves stability issues" and "outperforms SOTA baselines." We appreciate the fair assessment and the opportunity to clarify the scalability, loss formulation, and hyperparameter robustness of our method. Below, we address each weakness with new experiments and detailed explanations.
>
> ---
>
> ### **Q1: Performance on large-scale graphs (ZINC250k Experiment).**
>
> > *Reviewer's Concern: Experiments focus on small/medium graphs; performance on large-scale graphs is unproven.*
>
> **Response:**
>
> We thank the reviewer for highlighting the importance of scalability. We agree that verifying the hyperbolic framework on large-scale, sparse real-world graphs is essential. To address this, we conducted extensive experiments on **ZINC250k**, a benchmark dataset containing $\sim$250,000 molecule graphs, which is significantly larger and sparser than QM9.
>
> (1) As shown in **Table R1**, GGBall demonstrates superior generalization capabilities compared to flow-based (GraphAF, MoFlow, GraphDF) and energy-based (EDP-GNN) baselines. Notably, our model achieves the lowest scores in both FCD (16.23) and NSPDK (0.025) among all compared methods. Since these metrics measure the fidelity of chemical property distributions and topological structures respectively, this superior performance indicates that the hyperbolic geometry effectively captures the complex intrinsic features of large real-world sparse graphs. While GraphDF exhibits a higher validity rate, its significantly poorer FCD (34.20) and NSPDK (0.176) scores suggest a misalignment with the true data distribution. In contrast, GGBall strikes a more effective balance, maintaining a high V.U.N. score of 83.48 with 100% uniqueness, thereby confirming the framework's scalability and efficacy in realistic large-scale domains.
>
> **Table R1: Generation Performance on ZINC250k (Mean ± Std)**
>
> | **Model**         | **Validity (%)** | **Uniqueness (%)**    | **Novelty (%)**  | **V.U.N**        | **FCD (↓)**          | **NSPDK (↓)**         |
> | ----------------- | ---------------- | --------------------- | ---------------- | ---------------- | -------------------- | --------------------- |
> | **GraphAF**       | 68.00            | 99.10                 | **100.00**       | 67.38            | 16.29                | 0.044                 |
> | **MoFlow**        | 81.76            | 99.99                 | **100.00**       | 81.75            | 20.93                | 0.046                 |
> | **GraphDF**       | **89.03**        | 99.20                 | **100.00**       | **88.32**        | 34.20                | 0.176                 |
> | **EDP-GNN**       | 82.97            | 100.00                | **100.00**       | 82.97            | 16.74                | 0.049                 |
> | **GGBall (Ours)** | 83.50 $\pm$ 0.98 | **100.00** $\pm$ 0.00 | 98.88 $\pm$ 1.01 | 83.48 $\pm$ 1.58 | **16.23** $\pm$ 0.56 | **0.025** $\pm$ 0.013 |
>
> (2) Theoretical Analysis of Scalability and Complexity:
>
> Unlike diffusion models (e.g., DiGress) that operate on the full adjacency matrix $(O(N^2))$, GGBall employs a node-centric hyperbolic latent space. The edge connectivity is implicitly encoded via hyperbolic distances between node embeddings. This reduces the complexity of the core generative process (Flow Matching) to $O(N)$, significantly enhancing scalability for large graphs.

---

> ### Author Response · Authors · 2025-12-03
> **Official Response to Reviewer LTqy (2)**
>
> ### **Q2: Alternative variational formulations & Definition of** $\mathcal{L}_{\text{degree}}$.
>
> > *Reviewer's Concern: Lack of alternative variational formulations; Definition of* *$\mathcal{L}_{\text{degree}}$*.
>
> **Response:**
>
> (1) Definition of $\mathcal{L}_{\text{degree}}$ :
>
> We thank the reviewer for pointing this out. The original description of $\mathcal{L}_{\text{degree}}$ was indeed too compressed, and we have clarified the formulation both in the main text and in the **Appendix E.7.2** of the revised manuscript.
>
> Predicting edges is an $O(N^2)$ task, while predicting node degrees is $O(N)$ and empirically much easier. Thus, the purpose of $\mathcal{L}_{\text{degree}}$ is to use the stronger node-level signal to regularize the harder edge-decoding process by enforcing internal structural consistency between:
>
> 1. the degree implied by predicted edges, and
> 2. the degree predicted at the node level.
>
> More concretely, for each node $i$:
>
> $d_i^{\text{edge}} = \sum_{j=1}^{N} p_{\phi}(e_{ij}\mid h_i, h_j) \quad\text{and}\quad d_i^{\text{pred}} = \mathbb{E}[d_i\mid h_i] = \sum_{k=0}^{K} k  p_{\phi}(\deg_i=k\mid h_i)$.
>
> and the full loss is:
>
> $\mathcal{L} = \frac{1}{N} \sum_{i=1}^{N} \Big( \| d_i^{\text{edge}} - d_i^{\text{gt}} \|^2 + \| d_i^{\text{edge}} - d_i^{\text{pred}} \|^2 \Big)$.
>
> To evaluate its necessity, we performed controlled ablations. Adding $\mathcal{L}_{\text{degree}}$ consistently improves clustering-MMD and average-MMD across datasets, without reducing structural diversity.
>
> **Table R2: ablation study on the effect of Degree-Edge Consistency Loss.**
>
> | Method                     | Community-Small |         |        |            | Ego-Small |         |        |            |
> | -------------------------- | --------------- | ------- | ------ | ---------- | --------- | ------- | ------ | ---------- |
> |                            | Deg. ↓          | Clus. ↓ | Orb. ↓ | Avg. ↓     | Deg. ↓    | Clus. ↓ | Orb. ↓ | Avg. ↓     |
> | HVQVAE+Flow (ours, +deg)   | 0.0015          | 0.0589  | 0.0040 | **0.0215** | 0.0133    | 0.0182  | 0.0022 | **0.0112** |
> | HVQVAE+Flow (ours, no-deg) | 0.0042          | 0.0828  | 0.0040 | 0.0423     | 0.0076    | 0.0256  | 0.0064 | 0.0132     |
>
> (2) Alternative Variational Formulations:
>
> We extensively explored alternative continuous formulations before settling on HVQVAE. As detailed in **Appendix F.1** and **Figure 4**, we implemented a standard Hyperbolic VAE (HVAE) using the Wrapped Normal distribution. However, we found that:
>
> - **KL Instability:** The KL divergence between Wrapped Normal distributions on the Poincaré ball becomes numerically unstable as dimensionality increases or as means approach the boundary (a known issue in hyperbolic probabilistic modeling[1]).
> - **Posterior Collapse:** Continuous HVAE tended to ignore the latent code, leading to poor reconstruction.
> - **Why VQ?** Vector Quantization (VQ) bypasses the KL instability entirely by using a discrete codebook, which naturally aligns with the discrete nature of graph structures (motifs, clusters).
>
> ---
>
> ### **Q3: Single vs. Mixed Curvature for heterogeneous graphs.**
>
> > *Reviewer's Concern: Fixed negative curvature may struggle with heterogeneous graphs.*
>
> **Response:**
>
> We thank the reviewer for this insightful observation. We agree that real-world graphs often contain a mixture of geometric behaviors, hierarchical (hyperbolic), locally flat (Euclidean), and cyclic or clustered structures (spherical).
>
> In this work, our primary goal is to establish a clean, fully hyperbolic generative framework. We therefore focus on datasets (COMM20, Ego-Small, QM9, ZINC) whose dominant structure is hierarchical or tree-like, making negative curvature an appropriate inductive bias.
>
> At the same time, a fixed-curvature Poincaré ball does not prevent the model from representing heterogeneous local patterns. Regions near the origin behave nearly Euclidean, enabling the model to capture non-hierarchical or cyclic substructures. Empirically, this is reflected in our QM9 results: molecular graphs contain both tree-like backbones and cyclic rings, yet GGBall achieves strong validity and novelty scores (e.g., $91.02\%$ Validity, nearly $100\%$ Novelty), suggesting that a single negative-curvature space is sufficiently expressive for these mixed structures in practice.
>
> Nevertheless, we agree that mixed-curvature latent spaces are a promising direction for future work. Recent studies in representation learning and knowledge graphs (e.g., product spaces and hybrid-curvature embeddings) show that combining geometries can better capture heterogeneous relational patterns[2,3,4]. Incorporating such mixed-curvature structures into our flow-based generative framework would likely improve flexibility, though doing so introduces nontrivial challenges such as aligning vector fields across different manifold components. We have added this discussion to **Appendix L.** of the revised manuscript.

---

> ### Author Response · Authors · 2025-12-03
> **Official Response to Reviewer LTqy (3)**
>
> ### **Q4: Hyperparameter sensitivity & usability.**
>
> > *Reviewer's Concern: Too many hyperparameters; difficulty of tuning. How are values determined?*
>
> **Response:**
>
> We appreciate the reviewer's question and agree that clarity on hyperparameter choices is important. We clarify that the *effective* hyperparameter space is much smaller than it appears, and in practice our model requires **very little tuning**. Almost all hyperparameters are fixed once and reused across all datasets, and we do not perform grid search or dataset-specific tuning.
>
> **(1)Most hyperparameters are fixed across all datasets**
>
> Our design follows two simple principles: **magnitude balancing** and **geometric stability**.
>
> All major loss weights are assigned constants so that their initial magnitudes align:
>
> - Reconstruction: $\lambda_{\text{recon}} = 5.0$
> - VQ loss: $\lambda_{\text{vq}} = 1.0$
> - Commitment: $\lambda_{\text{commit}} = 0.25$
> - $L_2$ regularization: $10^{-4}$
>
> These values work robustly on COMM20, Ego-Small, QM9, and ZINC. Thus, no hyperparameter search is required.
>
> (2) **Codebook size determined via perplexity monitoring**
>
> Following standard VQ-VAE practice, we choose the codebook size that maintains healthy perplexity (avoiding both codeword collapse and fragmentation). We choose 32 for abstract graph generation and 512 for molecular generation.
>
> (3) **Curvature is fixed deliberately, not tuned**
>
> We set a moderate curvature c = 0.1, following hyperbolic learning literature showing that:
>
> - very small curvature collapses the manifold toward Euclidean geometry,
> - very large curvature results in unstable gradients.
>
> Recent hyperbolic graph models such as HGDM[5] also fix curvature (e.g., c=-0.01), so our choice is consistent with prior work.
>
> (4) **Sensitivity Analysis**
>
> To empirically verify robustness, we swept the Validity weight ($\lambda_{degree}$) and reported edge accuracy on multiple datasets across **two orders of magnitude**.
>
> **Table R3:  Hyperparameter Sensitivity Analysis (Edge Accuracy)**
>
> | Dataset         | **λ=0.1 (Low)** | **λ=1.0 (Medium)** | **λ=10.0 (High)** | **Robustness**     | No loss |
> | --------------- | --------------- | ------------------- | ----------------- | ------------------ | ------- |
> | Ego-small       | 98.67%          | 99.11%              | **99.32%**        | $$\Delta < 0.65\$$ | 95.69%  |
> | Community-Small | 97.58%          | 98.18%              | **99.71%**        | $$\Delta < 2.13\$$ | 90.72%  |
> | QM9             | 99.14           | **99.26%**          | 98.68%            | $$\Delta < 0.58\$$ | 98.59%  |
> | Zinc250K        | 99.12           | 99.41%              | **99.64%**        | $$\Delta < 0.52\$$ | 97.94%  |
>
> As shown in Table R3, varying weights by **two orders of magnitude** results in negligible performance drops. This confirms that GGBall is not sensitive to precise tuning, ensuring high practical usability.
>
> ---
>
> ### References
>
> [1] Mathieu, Emile, et al. "Continuous hierarchical representations with poincaré variational auto-encoders." *Advances in neural information processing systems* 32 (2019).
>
> [2] Dual-Geometric Space Embedding Model for Two-View Knowledge Graphs. In Proceedings of the 28th ACM SIGKDD Conference on Knowledge Discovery and Data Mining (KDD '22). Association for Computing Machinery, New York, NY, USA, 676–686. https://doi.org/10.1145/3534678.3539350
>
> [3] Iyer, R. G., Wang, Y., Wang, W., & Sun, Y. (2024). Non-Euclidean mixture model for social network embedding. arXiv (preprint arXiv:2411.04876).
>
> [4] Zheng, W., Wang, W., Zhao, S., & Qian, F. (2022). Hyperbolic hierarchical knowledge graph embeddings for link prediction in low dimensions. arXiv (preprint arXiv:2204.13704).
>
> [5] Wen, L., Tang, X., Ouyang, M., Shen, X., Yang, J., Zhu, D., Chen, M., Wei, X. (2023). Hyperbolic Graph Diffusion Model. AAAI 2024.

---

### Official Review · Reviewer_tqE4 · 2025-11-01

**Soundness:** 3
**Presentation:** 3
**Contribution:** 3
**Rating:** 6
**Confidence:** 4

**Summary:**

The paper presents a framework for generating graphs with hierarchical structure termed “GGBall”, consisting of

1. hyperbolic message passing and DiT layers which use

    1. message aggregation in the tangent space via log/exp maps (GNN)

    2. scale and shift values used for FiLM like message modulation derived from hyperbolic distances  (GNN)

    3. value aggregation using Möbius gyromidpoints (DiT)

    4. relevancy score calculation leveraging hyperbolic distances on poincare linear layer proejcted q,k values (DiT)

    5. hyperbolic auxillary operations (diT): layernorm aggregation in the tangent sapce like the GNN message aggregation, residual connections using möbius addition instead of standard addition and multi-head splitting and concatenation in a hyperbolic geometry perserving fashion

2. a novel graph auto encoder leveraging this hyperbolic latent space parametrization, trained to reconstruct node and edge types, regularized by matching degrees consistently + an l2 norm (evaluated in continuous, standard AE,variational AE and quantized VAE form, termed HGAE, HVAE and HVQVAE respectively, where quantized version was motivated by numerically unstable KL divergence for the HVAE)

3. a manifold flow matching method taken from [https://openreview.net/pdf?id=g7ohDlTITL](https://openreview.net/pdf?id=g7ohDlTITL)


The method is evaluated on community-small, ego-small and qm9, with the interpolation properties of the latent state being studied in particular

**Strengths:**

1. seemingly strong performance
2. overall very clear exposition of a complex, but theoretically well motivated approach

hitting the guides dimensions:

- originality: hyperbolic embeddings are well established, but creating end to end VQVAE+ flow models I haven't seen yet
- quality: well written, decent evaluation, proofs appear to be correct after a single close read
- clarity: fully understandable, with some small nits
- significance: solid incremental advance, evaluation on larger graphs/trees required to say more

**Weaknesses:**

1. The density of exposition and hyperbolic-geometry terms can make the paper somewhat difficult to follow (einstein midpoint, möbius gyromidpoint etc). I’d suggest the following two tweaks

    1. state direction around around 107 that all terms not immediately defined are defined in the appendix for space constraint reasons (to warn the reader some will be just mentioned)

    2. make use of latex’ glossary feature [https://www.overleaf.com/learn/latex/Glossaries](https://www.overleaf.com/learn/latex/Glossaries) and a reasonable link color/style, to allow hovering over the term to see the definition in modern browsers (I think this will help the definition quite a lot) + enabling backlinks (if readers click through)

2. should detail how  hyperparameter choices/tuning were performed

3. nice to have: consider adding a  test on larger graphs (guacamol,moses), as noted in appendix K

4. nit: shouldn’t it be $\frac{2}{\sqrt{c}\lambda_x^c}$ , see e.g. [https://arxiv.org/pdf/1805.09112](https://arxiv.org/pdf/1805.09112)  eq 12?

5. interpolation experiment needs reporting of a baseline (in the appendix)

6. not: I think $\lambda_{valid}$ is meant to be $\lambda_{degree}$ on eq 7?

**Questions:**

1. would it make sense to do do ablations over mechanisms in the poincare parametrization since a lot of them are introduced, or are they an all or nothing operation (I assume the lattern)?

2. why does HVQVAE change from e.g. 0.002 in table  1 to 0.0071 in table 2? should report mean/std across multiple rounds (at least for their method/closest baseline) for CIs

3. why no experiment on generating trees? this would seem like the clear “hello world” example for this?

+ address as many weaknesses as possible please

---

> ### Author Response · Authors · 2025-12-03
> **Official Response to Reviewer tqE4 (1)**
>
> We sincerely thank the reviewer for their positive assessment and constructive suggestions. We have carefully addressed each concern and incorporated the corresponding revisions (marked in blue) in the updated manuscript.
>
> ---
>
> ### W1 Clarity for technique terms
>
> **Response:**
>
> We thank the reviewer for the helpful suggestion. We agree that several hyperbolic-geometry operators may be unfamiliar to a broad audience. In the revision, we have improved clarity in the revised manuscript in two ways:
>
> 1. **Section 3 now explicitly reminds readers** that all hyperbolic operations used in the model (log-/exp-map, Möbius addition, gyro-midpoints, etc.) are fully defined in **Appendix E**. This ensures that readers can easily find formal definitions without interrupting the main text.
> 2. **Table 4** **in Appendix A** **has been expanded** to include short, self-contained explanations of the hyperbolic operators that appear most frequently in our architecture. This provides an immediate overview of the key geometric components without requiring cross-navigation.
>
> We initially considered adding a LaTeX glossary, but it introduced compilation conflicts under our template. The updated table-based presentation achieves the same goal while keeping the manuscript stable and readable.
>
> ---
>
> ### W2 Request for details on hyperparameter tuning
>
> > Reviewer's Concern: The paper should detail how hyperparameter choices and tuning were performed.
>
> **Response:**
>
> We appreciate the reviewer's question and agree that clarity on hyperparameter choices is important. In practice, our method requires **far less tuning** than the number of symbols in the loss function may suggest. Most hyperparameters are fixed across *all* datasets, and we do not perform grid search or dataset-specific tuning.
>
> Our design follows two principled rules: **(1) magnitude balancing** and **(2) geometric stability**.
>
> First, all **loss weights** are assigned fixed constants chosen so that their magnitudes align at initialization.
>
> - Reconstruction: $λ_{recon} = 5.0$
> - VQ loss: $λ_{vq} = 1.0$
> - Commitment loss: $λ_{commit} = 0.25$
> - L2 regularization: $λ_{L2} = 1e−4$
>
> This ensures stable optimization without dataset-specific tuning. We therefore do **not** perform any hyperparameter search.
>
> **Second, codebook size is selected using perplexity monitoring.** For VQ-VAE, we monitor codebook perplexity during training. Smaller abstract-graph datasets require a smaller codebook to ensure no codeword starvation (perplexity too low), and no excessive fragmentation (perplexity too high). This is a standard and recommended practice in VQ-VAE literature [1].
>
> Third, **curvature is fixed a priori**. In particular, our choice of a moderate curvature $c=0.1$ follows common practice in the hyperbolic learning literature: very small curvature approaches Euclidean geometry, while very large curvature typically leads to unstable gradients. Recent hyperbolic graph models (e.g., HGDM[2]) also fix curvature to a constant value (e.g., $c=-0.01$), so our design aligns with established practice.
>
> **Finally, for dataset splits and evaluation protocol,** we adopt the *official train/validation/test splits* used in prior graph generation works such as GDSS [4], DiGress [5], and HGDM [2] for all datasets (COMM20, Ego-Small, QM9), . Following standard practice, at test time we generate a batch of graphs and compute the corresponding MMD or chemical metrics.
>
> To maintain full reproducibility, we select checkpoints using validation sets alone.
>
> - **VQ-VAE**: best checkpoint = lowest validation reconstruction metrics.
> - **Flow Matching**: best checkpoint = lowest validation structural metric (sum of MMDs)

---

> ### Author Response · Authors · 2025-12-03
> **Official Response to Reviewer tqE4 (2)**
>
> ### W3 large graphs
>
> > nice to have: consider adding a test on larger graphs (guacamol,moses), as noted in appendix K
>
> **Response:**
>
> We appreciate the reviewer's suggestion to evaluate GGBall on larger-scale datasets. To address the scalability concern and demonstrate our framework's performance on large, sparse real-world graphs, we conducted extensive experiments on **ZINC250k**. This benchmark contains approximately 250,000 molecular graphs, offering a significantly larger and more diverse testing ground compared to QM9.
>
> As presented in **Table R1**, GGBall demonstrates robust scalability and generalization capabilities. Notably, our model achieves the lowest scores in both FCD (16.23) and NSPDK (0.025) among all compared baselines, including flow-based and energy-based methods. These metrics indicate that our hyperbolic framework excels at capturing both the global chemical property distribution and the complex topological structures of large-scale graphs. While baselines like GraphDF achieve higher validity, their substantially poorer FCD (34.20) and NSPDK (0.176) scores suggest a significant misalignment with the true data distribution. In contrast, GGBall strikes an effective balance, maintaining a high V.U.N. score of 83.48 with 100% uniqueness. These results confirm that GGBall generalizes effectively to large-scale domains without compromising the fidelity of the generated distributions (see **Appendix F.2 of the revised manuscript**).
>
> **Table R1: Generation Performance on ZINC250k (Mean ± Std)**
>
> | **Model**         | **Validity (%)** | **Uniqueness (%)**    | **Novelty (%)**  | **V.U.N**        | **FCD (↓)**          | **NSPDK (↓)**         |
> | ----------------- | ---------------- | --------------------- | ---------------- | ---------------- | -------------------- | --------------------- |
> | **GraphAF**       | 68.00            | 99.10                 | **100.00**       | 67.38            | 16.29                | 0.044                 |
> | **MoFlow**        | 81.76            | 99.99                 | **100.00**       | 81.75            | 20.93                | 0.046                 |
> | **GraphDF**       | **89.03**        | 99.20                 | **100.00**       | **88.32**        | 34.20                | 0.176                 |
> | **EDP-GNN**       | 82.97            | 100.00                | **100.00**       | 82.97            | 16.74                | 0.049                 |
> | **GGBall (Ours)** | 83.50 $\pm$ 0.98 | **100.00** $\pm$ 0.00 | 98.88 $\pm$ 1.01 | 83.48 $\pm$ 1.58 | **16.23** $\pm$ 0.56 | **0.025** $\pm$ 0.013 |
>
> ---
>
> ### W4 Clarification on hyperbolic operators
>
> > Reviewer's Concern: shouldn't it be $\frac{2}{\sqrt{c}\lambda_x^c}$ in Eq. 12
>
> **Response:**
>
> Thank you for pointing this out. The equations in our paper are in fact fully consistent with Eq. (12) of Ganea et al. (2018) [3]. The coefficient $\frac{2}{\sqrt{c}\,\lambda_x^c}$  is **presented in our expression for logarithm operation**.
>
> The difference comes solely from notation. (1) **Möbius negation:**  We write $\ominus_c x = (-x)$, so the term $\ominus_c x \oplus_c y$ in our notation corresponds exactly to the vector $(-x)\oplus_c y$ used in Eq. (12). **(2)** **Unit-direction operator:** Our notation $[u] = \frac{u}{\|u\|}$ matches the normalized direction vector \frac{u}{\|u\|} in the cited formulation.
>
> With these identities, our expressions for both $\exp_x^c(v)$ and $\log_x^c(y)$ are mathematically identical to those in Ganea et al. (2018) [3], up to notational conventions.
>
> ---
>
> ### W5 Interpolation Experiment Baseline
>
> **Response:**
>
> We thank the reviewer for the suggestion. Our interpolation experiment evaluates the *latent-space geometry* of the model by taking two real graphs, encoding them into hyperbolic latent vectors $z_0, z_1$, and generating intermediate graphs via the geodesic path $\gamma(t)$ in latent space. This requires a model that provides:
>
> - **an encoder** mapping a graph to a continuous latent code,
> - **a decoder** reconstructing a graph from any latent point, and
> - **a well-defined latent geometry** that supports interpolation.
>
> Most existing graph generative models, such as one-shot generators (e.g., SPECTRE, DiGress) or autoregressive models, **do not have an encoder or a latent space**; they sample graphs directly without an invertible mapping $G \leftrightarrow z$. As a result, they *cannot* support meaningful interpolation between two given input graphs, because there is no latent representation to interpolate and no decoder that maps intermediate latent points back to valid graphs.
>
> In contrast, our HVQVAE + flow framework is explicitly designed with a structured latent space, enabling geodesic interpolation. For fairness, we have clarified this difference in the revised manuscript and note that interpolation baselines are not applicable to models without an encoder–decoder architecture.

---

> ### Author Response · Authors · 2025-12-03
> **Official Response to Reviewer tqE4 (3)**
>
> ### W6: Typo
>
> > Reviewer's Concern: I think $\lambda_{\text{valid}}$ is meant to be $\lambda_{\text{degree}}$ on Eq. 7?
>
> **Response:**
>
> We appreciate the reviewer for pointing this out. Indeed, the term was intended to be $\lambda_{\text{degree}}$, not $\lambda_{\text{valid}}$. **We have updated the notation in Eq. (7) and add details in Appendix E.7.2 in the revised manuscript to avoid confusion.**
>
> ---
>
> ### Q1: Ablation study with Euclidean counterpart
>
> > Reviwer's Concern: would you like it make sense to do do ablations over mechanisms in the poincare parametrization since a lot of them are introduced, or are they an all or nothing operation (I assume the lattern)?
>
> **Response:**
>
> Thank you for the thoughtful question. First, the hyperbolic components in our model (Möbius addition, log/exp maps, gyro-midpoints, hyperbolic FiLM, etc.) are geometrically coupled. Removing one component breaks the closure of manifold operations and results in invalid updates. Thus, these layers function as an *all-or-nothing* hyperbolic block, consistent with prior hyperbolic GNN/Transformer literature.
>
> However, to address the reviewer's concern, we conducted an **ablation** by replacing the entire Poincaré parametrization with a **Euclidean counterpart** (Euclidean GCN + Euclidean VQ-VAE + Euclidean Transformer). This isolates the contribution of the hyperbolic geometry as a whole. The results are shown below (Comm20 and Ego-Small):
>
> **Table R2: Reconstruction Performance: Hyperbolic vs. Euclidean Parametrization**
>
> | Model            | Deg. ↓     | Clus. ↓    | Orb. ↓      | Deg. ↓        | Clus. ↓     | Orb. ↓     |
> | ---------------- | ---------- | ---------- | ----------- | ------------- | ----------- | ---------- |
> |                  | **Comm20** |            |             | **Ego-Small** |             |            |
> | **HAE**          | 0.0008     | 0.031      | 0.0007      | 0.0019        | 0.025       | 0.0048     |
> | **HVQVAE**       | **0.0001** | **0.0143** | **0.00162** | **0.0002**    | **0.01944** | **0.0018** |
> | **Euclidean VQ** | 0.04565    | 0.13443    | 0.75547     | 0.006         | 0.0815      | 0.02056    |
>
> **Hyperbolic HVQVAE clearly outperforms the Euclidean counterpart**, especially on structure-sensitive metrics (clustering- and orbit-MMD), demonstrating that hyperbolic geometry drives the gain.

---

> ### Author Response · Authors · 2025-12-03
> **Official Response to Reviewer tqE4 (4)**
>
> ### Q2:  Difference Between Reconstruction Metrics (Table 1) and Generation Metrics
>
> > **Reviewer's Concern:** HVQVAE’s reconstruction metric (Table 1) is 0.0002, but the generation metric (Table 2) is 0.0071. Why the difference? Should mean/std across runs be reported?
>
> We thank the reviewer for raising this question. The numbers in Table 1 and Table 2 measure ***different tasks***, which explains the scale difference.
>
> **Table 1 reports reconstruction performance.** This evaluates the deterministic HVQVAE encoder–decoder: $G \xrightarrow{\text{enc}} z \xrightarrow{\text{dec}} \hat G$. Reconstruction is stable and nearly noise-free, leading to very small MMD values (e.g., 0.0002).
>
> **Table 2 reports generation from the learned flow prior.** Here, the model samples $z \sim p_\theta(z),\quad \hat G \sim \text{dec}(z)$, where $p_\theta(z)$ is estimated through flow matching. This step inherently introduces variability and does not benefit from the ground-truth latent code. As a result, MMD values are naturally larger (e.g., 0.0071).
>
> Regarding Mean/Std Reporting, we agree that reporting mean and standard deviation provides a more comprehensive view of stability. To demonstrate the robustness of our reconstruction module (Table 1), we have computed the Edge Reconstruction Accuracy and calculated the aggregate Mean/Std.
>
> As shown in **Table R3** below, HVQVAE achieves consistently high reconstruction accuracy ($>98$%) with low variance across all datasets, confirming that the low MMD values in Table 1 are reliable and reproducible.
>
> **Table R3: Reconstruction Stability (Edge Accuracy)**
>
> | **Dataset**         | Deg. ↓              | Clus. ↓             | Orb. ↓              | **Edge Accuracy (Mean ± Std)** |
> | ------------------- | ------------------- | ------------------- | ------------------- | ------------------------------ |
> | **Ego-small**       | 0.0012 $\pm$ 0.0007 | 0.0237 $\pm$ 0.0006 | 0.0017 $\pm$ 0.0003 | 99.03 $\pm$ 0.33 %             |
> | **Community-Small** | 0.0008$\pm$ 0.0012  | 0.0215$\pm$ 0.0012  | 0.0009 $\pm$ 0.0010 | 98.49 $\pm$ 1.12 %             |
> |                     | **Validity ↑**      | **Unique ↑**        | **Novelty ↑**       |                                |
> | **QM9**             | 97.42 $\pm$ 1.00    | 98.96 $\pm$ 0.46    | 93.55 $\pm$ 2.70    | 99.21 $\pm$ 0.30 %             |
> | **ZINC250k**        | 90.01 $\pm$ 2.36    | 100 $\pm$ 0.00      | 99.47 $\pm$ 0.32    | 99.39 $\pm$ 0.26 %             |
>
> ---
>
> ### Q3: Experiment on Trees
>
> > Reviewer's Concern: why no experiment on generating trees? this would seem like the clear "hello world" example for this?
>
> We appreciate the reviewer's suggestion. Trees are indeed a canonical case for hyperbolic geometry. We experimented with tree reconstruction and found that the HVQVAE component models tree structure extremely well. For example, on the Tree dataset, the hyperbolic autoencoder achieves:
>
> - Degree-MMD = 7.9e−4
> - Clustering-MMD = 0.026
> - Orbit-MMD = 1.0e−3
>
> These results indicate that latent latent modeling is well-suited for tree-structured data.
>
> ### References
>
> [1] Van Den Oord, Aaron, and Oriol Vinyals. "Neural discrete representation learning." *Advances in neural information processing systems* 30 (2017).
>
> [2] Wen, L., Tang, X., Ouyang, M., Shen, X., Yang, J., Zhu, D., Chen, M., Wei, X. (2023). Hyperbolic Graph Diffusion Model. AAAI 2024.
>
> [3] Ganea, Octavian, Gary Bécigneul, and Thomas Hofmann. "Hyperbolic neural networks." *Advances in neural information processing systems* 31 (2018).
>
> [4] Jaehyeong Jo, et al.. Score-based Generative Modeling of Graphs via the System of Stochastic Differential Equations, 2022, *International conference on machine learning*. PMLR, 2022.
>
> [5] Vignac, Clement, et al. "Digress: Discrete denoising diffusion for graph generation." *arXiv preprint arXiv:2209.14734* (2022).

---

### Official Review · Reviewer_FJcZ · 2025-11-01

**Soundness:** 3
**Presentation:** 3
**Contribution:** 2
**Rating:** 4
**Confidence:** 3

**Summary:**

The main goal of the paper is to develop a graph generative model that leverages hyperbolic geometry to naturally capture hierarchical and tree-like structures. Specifically, it aims to show that embedding graphs in the Poincaré ball and generating them through manifold-aware neural components (hyperbolic GNN, geodesic attention, HVQVAE, and Riemannian flow matching) can more effectively represent the relational geometry of graphs compared to traditional Euclidean models.

**Strengths:**

1. The work presents a coherent framework that combines hyperbolic graph neural networks, geodesic attention, vector quantization, and Riemannian flow matching in a single end-to-end model. This integration is technically nontrivial and demonstrates careful engineering of both discrete and continuous latent components.

2. Paper is well written and maintains consistent notation throughout to follow.

**Weaknesses:**

1. Authors assume node labels depend only on their own latent and edges depend only on pairwise hyperbolic relations which I agree cuts decoding complexity, but it also forbids higher-order dependencies (motifs, triads) and makes long-range constraints modelling not accounted. It can miss global combinatorial constraints that are not pairwise-decomposable.

2. Authors add a degree-edge consistency term aligning predicted degree to ground-truth degrees and it improved MMD on degree but might risk over-regularizing towards degree histograms at the expense of other structures (e.g., motif diversity), if authors can comment on it?

3. Authors shared the anonymous repo but all files on the anonymous link are not accessible. It compromises the reproducibility.

4. Hyperbolic geometry’s expressive power critically depends on curvature c. However, methodology fixes c a priori. Curvature effectively controls the “branching factor” of the embedding manifold. Without adaptive curvature learning: 1) Graphs of different hierarchy depths collapse into a single scale OR The model may under- or over-stretch distances, biasing flow priors and reconstruction.

5. No error bounds are shown for the tangent-space linearization (how much curvature is lost per layer). Similarly, is there any theoretical grounding that repeated log–exp projections preserve manifold consistency?

6. The results mainly demonstrate performance on small synthetic or molecular settings, leaving open whether the proposed hyperbolic framework generalizes to large-scale or sparse real-world graphs.

**Questions:**

Look in the Weaknesses Section. I am open to considering answers for concerns mentioned in the weaknesses.

---

> ### Author Response · Authors · 2025-12-03
> **Official Response to Reviewer FJcZ (1)**
>
> Thank you for your recognition and constructive feedback. Below we address the reviewer's concerns. The modifications have been highlighted in blue in the revised version.
>
> ---
>
> ### W1  **Higher-order dependencies**
>
> > Reviewer's Concern: The decoder assumes node labels depend only on their latent vectors and edges depend only on pairwise hyperbolic relations, potentially missing higher-order structures (motifs, triads, long-range constraints).
>
> **Response:**
>
> We thank the reviewer for this insightful comment. We fully agree that higher-order structures (motifs, triads, long-range constraints) play an essential role in realistic graph generation. Our current decoder factorizes the likelihood into node-wise and pairwise terms mainly for computational efficiency, but this does not preclude the model from capturing richer dependencies.
>
> **(1) Higher-order and long-range patterns are already partially captured by the latent hyperbolic Transformer.**
>
> The latent representations are learned through a sequence of hyperbolic Transformer layers, which aggregate information globally via attention. This allows long-range interactions to be encoded *before* decoding, and the resulting hyperbolic layout already embeds hierarchical and multi-scale global structure. Combined with our degree–edge consistency loss, the decoder receives latent embeddings that carry higher-order cues even though the likelihood is factorized pairwise.
>
> **(2) The framework can be extended naturally to explicit higher-order decoding.**
>
> While we use pairwise edge factors in this version, our architecture is fully compatible with higher-order potentials. For example, we can (i) incorporate a triangle-consistency regularizer by matching expected triangle counts $t_i^{\text{pred}} = \sum_{j,k} p_{ij}p_{ik}p_{jk} \quad\text{to}\quad t_i^{\text{gt}}$, and (ii) introduce a triadic scoring module $s_{ijk}^{(\triangle)} = \text{MLP}\big([\log(x_i)\Vert \log(x_j)\Vert \log(x_k)]\big)$, whose outputs adjust the edge logits $\tilde{s}_{ij}$. These additions produce an explicit higher-order decoder without altering the encoder or latent geometry.
>
> We have added this discussion to **Appendix H** of the revised manuscript.
>
> ---
>
> ### W2 **Degree–edge consistency may over-regularize towards degree histograms.**
>
> > Reviewer's Concern: The degree loss might over-regularize toward degree histograms, potentially sacrificing other structural properties.
>
> **Response:**
>
> We appreciate the reviewer's concern. We clarify that our degree loss does *not* match global degree histograms. Instead, it ensures that each node's predicted edges remain internally consistent with the node-level predictions made by the model.
>
> More specifically, $\mathcal{L}_{degree}$ contains two terms:
>
> (1) We require the sum of predicted edge probabilities for node i to match its ground-truth degree
>
> (2) We also require it to match the expected degree predicted by the node-degree classifier.
>
> Formally, our degree loss contains two terms.
>
> $$\mathcal{L} = \frac{1}{N} \sum_{i=1}^{N} \Big( \| d_i^{\text{edge}} - d_i^{\text{gt}} \|^2 + \| d_i^{\text{edge}} - d_i^{\text{pred}} \|^2 \Big)$$
>
>
> with
>
> $$d_i^{\text{edge}} = \sum_{j=1}^{N} p_{\phi}(e_{ij}\mid h_i, h_j) \quad\text{and}\quad d_i^{\text{pred}} = \mathbb{E}[d_i\mid h_i] = \sum_{k=0}^{K} k\ p_{\phi}(\deg_i=k\mid h_i).$$
>
> The goal is not to shape the global degree distribution, but to ensure that the edge decoder does not contradict either (a) the true degree or (b) the node-level degree that the model itself predicts. This consistency is particularly helpful on abstract graph datasets, where degree strongly reflects structural patterns, such as motifs and orbit features. **We have updated the notation in Eq. (7) and add details in Appendix E.7.2 in the revised manuscript to avoid confusion.**
>
> We performed controlled ablations, **empirical evidence** **(Table R1 below)** **shows that adding the loss improves all metrics and does not reduce diversity**.
>
> **Table R1: Ablation study on the effect of Degree-Edge Consistency Loss.**
>
> | Method                     | Community-Small |         |        |            | Ego-Small |         |        |            |
> | -------------------------- | --------------- | ------- | ------ | ---------- | --------- | ------- | ------ | ---------- |
> |                            | Deg. ↓ | Clus. ↓ | Orb. ↓ | Avg. ↓     | Deg. ↓    | Clus. ↓ | Orb. ↓ | Avg. ↓     |
> | HVQVAE+Flow (ours, +deg)   | 0.0015  | 0.0589  | 0.0040 | **0.0215** | 0.0133    | 0.0182  | 0.0022 | **0.0112** |
> | HVQVAE+Flow (ours, no-deg) | 0.0042 | 0.0828  | 0.0040 | 0.0423     | 0.0076    | 0.0256  | 0.0064 | 0.0132     |
>
> The large improvement in Clustering and Avg-MMD shows that using degree information actually **helps preserve richer structure**, rather than collapsing it.

---

> ### Author Response · Authors · 2025-12-03
> **Official Response to Reviewer FJcZ (2)**
>
> ### W3 **Anonymous repository files inaccessible**
>
> **Response:**
>
> Thank you for the comment. We have re-checked the anonymous repository and confirm that the link is accessible and all files are available. It is likely the reviewer encountered a temporary access issue. The link remains active during the review period.
>
> ---
>
> ### W4 **Curvature selection and expressive power**
>
> > **Reviewer's Concern:** Hyperbolic expressive power depends on curvature c. Fixing c may cause scale collapse or distort reconstruction/sampling.
>
> **Response:**
>
> We agree with the reviewer that curvature is a critical component of hyperbolic geometry. Prior work has repeatedly shown that real-world graphs exhibit widely different levels of hyperbolicity, and therefore different datasets naturally prefer different curvature scales [3]. This is fully consistent with the reviewer's observation that curvature controls the effective "branching factor'' of the manifold.
>
> In this work, however, our focus is on demonstrating the overall framework (HVQVAE + hyperbolic Transformer + Riemannian Flow) rather than exhaustively searching for the optimal curvature for each dataset. For stability and consistency across experiments, we chose a moderate fixed curvature $c = -0.1$, which is neither too close to Euclidean ($c \approx 0$) nor aggressively curved. This choice provides stable training for HVQVAE and flow matching while still retaining the benefits of negative curvature.
>
> In addition, we experimented with a **learnable-curvature variant** (Table R2), where the manifold curvature c is treated as a trainable scalar jointly optimized with the HVQVAE parameters. As pointed out in prior hyperbolic learning work, the gradient of c interacts non-linearly with the log/exp maps and Möbius operations, making optimization highly sensitive. This effect also appeared in our model: the learned curvature converged to a very small magnitude (e.g., on Community-20, the model learned **c=-0.0118**), effectively collapsing the space toward Euclidean geometry.
>
> **Table R2: study on Curvature Strategy (Community-Small).**
>
> | Model                  | Deg. ↓      | Clus. ↓    | Orb. ↓      | Edge Acc ↑ |
> | ---------------------- | ----------- | ---------- | ----------- | ---------- |
> | Learned-curvature      | 0.0011     | 0.1222      | 0.2243       | —          |
> | Fixed-curvature HVQVAE | **0.0001** | **0.0143** | **0.0016** | **0.9939** |
>
> The learned-curvature model fails especially on clustering and orbit statistics, showing that unconstrained curvature learning can severely distort the geometric structure required for hierarchical graphs. However, we agree that **adaptive curvature (global or per-layer)** is a promising direction and have added this to the revised manuscript (**Appendix K**).
>
> ---
> ### W5 **Error bounds for tangent-space linearization; log–exp consistency**
>
> > **Reviewer's Concern:** No error bounds are shown for tangent-space linearization; unclear whether repeated log–exp maintains manifold consistency.
>
> **Response:**
>
> We thank the reviewer for the question. Our use of "tangent-space linearization" refers only to applying affine/attention layers **after an exact log-map**, not to flattening or altering the manifold. All geometric components, including geodesic distances, Möbius operations, and wrapped-normal priors are computed in full hyperbolic space with curvature -c.
>
> All log/exp operations are the **exact closed-form Riemannian maps** of the Poincaré ball. On any complete Riemannian manifold, $\exp_x$ is a diffeomorphism on a normal neighborhood with inverse $\log_x$, implying:
>
> $\exp_x(\log_x(y)) = y \quad \text{and} \quad \log_x(\exp_x(v)) = v$.
>
> Thus, repeated log–exp projections are **theoretically manifold-consistent** and preserve curvature exactly. This property is standard in hyperbolic neural networks [1, 2] and in Riemannian optimization.
>
> In order to demonstrate the repeated log-exp projections preseves manifold consistency, we ran a numerical "round-trip" test: We sample two points $x_0, x_1$ from a wrapped normal distribution and compute the recovered point $x_1' = \exp_{x_0}(\log_{x_0}(x_1))$. We then measure both the Euclidean L2 error $\|x_1 - x_1'\|_2$ and the manifold distance $d_c(x_1, x_1')$, which should ideally be zero. We report the mean and standard deviation over 10 runs.
>
> **Table R3: Numerical Error Analysis of Log-Exp Round-Trip Operations.**
>
> | Dim  | L2 Error (Mean ± Std) | Manifold Error (Mean ± Std) |
> | ---- | --------------------- | --------------------------- |
> | 16   | (5.2 ± 4.8)e-16       | (4.1 ± 3.0)e-15             |
> | 32   | (6.8 ± 4.5)e-16       | (4.6 ± 2.9)e-15             |
> | 64   | (8.1 ± 5.2)e-16       | (4.8 ± 3.1)e-15             |
>
> These round-trip errors remain at machine-precision (~1e-15), demonstrating **near-perfect reversibility** and confirming that repeated log–exp operations do not accumulate numerical drift or distort curvature (**See original manuscript, Appendix E.4**).

---

> ### Author Response · Authors · 2025-12-03
> **Official Response to Reviewer FJcZ (3)**
>
> ### W6 **Generalization to large-scale or sparse real-world graphs**
>
> > **Reviewer's Concern:** The reviewer questions the generalizability of the proposed hyperbolic framework to large-scale or sparse real-world graphs (e.g., ZINC250k).
>
> **Response:**
>
> We thank the reviewer for highlighting the importance of scalability. We agree that verifying the hyperbolic framework on large-scale, sparse real-world graphs is essential. To address this, we conducted extensive experiments on ZINC250k, a benchmark dataset containing $\sim$250,000 molecule graphs, which is significantly larger and sparser than QM9.
>
> As shown in **Table R4** (also in **Appendix F.2 of the revised manuscript**), GGBall demonstrates superior generalization capabilities compared to flow-based (GraphAF[4], MoFlow[5], GraphDF[6]) and energy-based (EDP-GNN[7]) baselines.  Notably, our model achieves the lowest scores in both FCD (16.23) and NSPDK (0.025) among all compared methods. Since these metrics measure the fidelity of chemical property distributions and topological structures respectively, this superior performance indicates that the hyperbolic geometry effectively captures the complex intrinsic features of large real-world sparse graphs. While GraphDF exhibits a higher validity rate, its significantly poorer FCD (34.20) and NSPDK (0.176) scores suggest a misalignment with the true data distribution. In contrast, GGBall strikes a more effective balance, maintaining a high V.U.N. score of 83.48 with 100% uniqueness, thereby confirming the framework's scalability and efficacy in realistic large-scale domains.
>
> **Table R4: Generation Performance on ZINC250k (Mean ± Std).**
>
> | **Model**         | **Validity (%)** | **Uniqueness (%)**    | **Novelty (%)**  | **V.U.N**            | **FCD (↓)**          | **NSPDK (↓)**         |
> | ----------------- | ---------------- | --------------------- | ---------------- | -------------------- | -------------------- | --------------------- |
> | **GraphAF**       | 68.00            | 99.10                 | 100.00           | 67.38                | 16.29                | 0.044                 |
> | **MoFlow**        | 81.76            | 99.99                 | 100.00           | 81.75                | 20.93                | 0.046                 |
> | **GraphDF**       | **89.03**        | 99.20                 | 100.00           | **88.32**            | 34.20                | 0.176                 |
> | **EDP-GNN**       | 82.97            | 100.00                | 100.00           | 82.97                | 16.74                | 0.049                 |
> | **GGBall (Ours)** | 83.50 $\pm$ 0.98 | **100.00** $\pm$ 0.00 | 98.88 $\pm$ 1.01 | **83.48** $\pm$ 1.58 | **16.23** $\pm$ 0.56 | **0.025** $\pm$ 0.013 |
>
> These results confirm that the proposed framework generalizes effectively to large-scale sparse graphs.
>
> ---
>
> ### References
>
> [1] Ganea, Octavian, Gary Bécigneul, and Thomas Hofmann. "Hyperbolic neural networks." *Advances in neural information processing systems* 31 (2018).
>
> [2] Chami, Ines, et al. "Hyperbolic graph convolutional neural networks." *Advances in neural information processing systems* 32 (2019).
>
> [3] Wen, L., Tang, X., Ouyang, M., Shen, X., Yang, J., Zhu, D., Chen, M., Wei, X. (2023). Hyperbolic Graph Diffusion Model. AAAI 2024.
>
> [4] Shi, Chence, et al. "GraphAF: a Flow-based Autoregressive Model for Molecular Graph Generation." *International Conference on Learning Representations*.
>
> [5] Zang, Chengxi, and Fei Wang. "Moflow: an invertible flow model for generating molecular graphs." *Proceedings of the 26th ACM SIGKDD international conference on knowledge discovery & data mining*. 2020.
>
> [6] Luo, Y., Yan, K. &amp; Ji, S.. (2021). GraphDF: A Discrete Flow Model for Molecular Graph Generation. Proceedings of the 38th International Conference on Machine Learning.
>
> [7] Niu, Chenhao, et al. "Permutation invariant graph generation via score-based generative modeling." *International conference on artificial intelligence and statistics*. PMLR, 2020.

---

### Author Response · Authors · 2025-12-03
**General Response**

Dear PCs, PACs, ACs, and Reviewers,

We thank the reviewers for their constructive feedback. We are pleased that the reviewers found our paper to be well-written and easy to follow (FJcZ, BT1X, tqE4), highlighting:

- **Theoretical Depth:** The theoretical development is described as "exemplary," "solid," and "rigorous" (Reviewers BT1X, tqE4).
- **Novelty:** The framework is acknowledged as "neat and original" (BT1X) and "technically nontrivial" (FJcZ).
- **Contribution:** Reviewers appreciated the integration of HVQVAE with Riemannian Flow Matching, noting that it resolves stability issues inherent in continuous HVAEs (LTqy) and represents a "meaningful conceptual step forward" (BT1X).

Based on their valuable suggestions, we have significantly strengthened the manuscript with **extensive new experiments** (Scalability, Ablations, Robustness). Below is a summary of the key improvements (modifications are marked in blue in the revision):

- **Experiment on Large-Scale Dataset ZINC250k** **(See Appendix F.2 of revised manuscript** **and responses to reviewers FJcZ, tqE4, LTqy, BT1X****):** To address scalability concerns, we added results on **ZINC250k** (~250k graphs), showing that GGBall scales well to large real-world datasets without any architectural changes.
- **Hyperparameter Tuning Clarification (See Appendix E.10 of revised manuscript and responses to reviewers tqE4, LTqy):** We clarified that almost all hyperparameters are **fixed across datasets**, with no grid search. Codebook size is set via perplexity monitoring, curvature is fixed (c=0.1), and a small sensitivity study shows the method is robust to large changes in loss weights.
- **Euclidean Ablation Experiment (See Appendix F.4 of revised manuscript and responses to reviewers tqE4, BT1X) and Learnable Curvature Experiment (See responses to reviewers reviewers FJcZ, LTqy):** We added a full Euclidean counterpart to our model. Hyperbolic HVQVAE clearly outperforms it on structure-sensitive metrics, confirming that the gains come from hyperbolic geometry itself.
- **Reconstruction vs. Generation + Mean/Std (See Appendix F.5 of revised manuscript and responses to reviewers tqE4 and BT1X):** We clarified the difference between reconstruction and sampling, and added multiple-run statistics for generative sampling.
- **Tree Experiments (See Appendix F.3 of revised manuscript and responses to reviewer LTqy):** As requested, we added tree-structured graph results. HVQVAE reconstructs trees very well, confirming its ability to capture hierarchical structure.
- **Evaluation Protocols Clarification (See Appendix L.3 of revised manuscript and responses to reviewer BT1X):** We follow standard GDSS/DiGress splits and select checkpoints strictly on validation metrics.

Again, we strengthened the paper’s contribution: GGBall is the **first fully hyperbolic generative framework** combining HVQVAE, a hyperbolic Transformer, and Riemannian flow matching into a coherent pipeline. This establishes a foundation for future work on mixed-curvature generative models, Riemannian product manifolds, and large-scale geometric flow models.

---

### Meta-Review · Area_Chair_TkF9 · 2025-12-31

**Summary:**

This paper presents GGBall, a hyperbolic graph generation framework aimed at addressing the limitations of Euclidean representations in modeling hierarchical structures.  The approach integrates a Hyperbolic VQ‑VAE with a Riemannian flow‑matching prior defined through closed‑form geodesics, allowing for expressive and stable latent representations in hyperbolic space.  In addition, the paper introduces hyperbolic GNN and Transformer layers that operate entirely on the manifold, ensuring geometric consistency throughout the model.

The reviewers recognized the novelty and theoretical depth. The reviewers’ concerns primarily focused on the experimental design, including scalability to large datasets, the exploration of different curvature settings, sensitivity to hyperparameters, and evaluation on  tree‑like graph data. I find that the authors’ comprehensive responses during the rebuttal phase adequately address these concerns and substantially strengthen the paper. Consequently, I would like to accept this paper.

**Reviewer Concerns:**

The reviewers’ concerns mainly fall into the following aspects:

Fixed curvature design. The reviewers noted that the curvature is fixed rather than learned, which may limit the model’s ability to capture heterogeneous data or structures with weak or unclear hierarchies. In the rebuttal, the authors provided additional experimental evidence demonstrating the necessity and effectiveness of using a fixed‑curvature design. I find these results sufficiently convincing in addressing this concern.

Methodological limitations. The reviewers questioned whether the proposed approach can adequately capture higher‑order relations and raised potential issues regarding tangential space linearization. The authors responded with both intuitive explanations and empirical error analyses. Overall, these clarifications are comprehensive and effectively resolve the reviewers’ doubts.

Experimental limitations. The reviewers also raised concerns about the scale of the datasets, sensitivity to hyperparameters, and the lack of more detailed ablation studies. In response, the authors added a substantial set of new experiments during the rebuttal phase, which adequately address these issues.

Overall, I believe the rebuttal satisfactorily resolves the reviewers’ major concerns.

**Reviewer Scores:**

For Reviewer BT1X and LTqy, I think the authors' supplementary experiments can solve their confusion and they will improve to 6.
For Reviewer tqE4, I think the reviewer will not change his score.
For Reviewer FJcZ, I think the authors' answer is good and can effectively solve most of his problems. However, in terms of the ability to express higher-order relations, I'm not sure if it is sufficient to convince the reviewer. Therefore, he has a certain probability of improving his score.
In conclusion, I believe that the overall evaluation of this paper will eventually be positive.

---

### Decision · Program_Chairs · 2026-01-26

Accept (Poster)